# Pruning Close to Home:
# Distance from Initialization impacts Lottery Tickets

## Abstract

The Lottery Ticket Hypothesis (LTH) states that there exist sparse subnetworks (called 'winning' Lottery Tickets) within dense randomly initialized networks that, when trained under the same regime, achieve similar or better validation accuracy as the dense network. It has been shown that for larger networks and more complex datasets, these Lottery Tickets cannot be found in randomly initializations, but that they require lightly pretrained weights. More specifically, the pretrained weights need to be stable to SGD noise, but calculating this metric involves an expensive procedure. In this paper, we take a closer look at certain training hyperparameters that influence SGD noise throughout optimization. We show that by smart hyperparameter selection we can forego the pretraining step and still find winning tickets in various settings. We term these hyperparameters early-stable, as networks trained with those become stable to SGD noise early during training, and discover that the tickets they produce, exhibit remarkable generalization properties. Finally, we hypothesize that a larger Learning Distance negatively impacts generalization of the resulting sparse network when iterative pruning, and devise an experiment to show this.

## 1 Introduction

Whenever a neural network model is trained, the typical approach is to start with a dense overparameterized neural network. This technique has proven to lead to the best results as it allows the model to converge to a global optimum in the loss landscape when trained under Gradient Descent, rather than a local optimum (Du et al., 2019). In contrast, starting training from a randomly sparse neural network typically fails to reach the same performance as an equivalent dense network (Li et al., 2016).

With the introduction of the Lottery Ticket Hypothesis (LTH) (Frankle & Carbin, 2019), the notion that sparse networks cannot reach the same performance as a dense network when trained from scratch has been shattered. By using a costly procedure called Iterative Magnitude Pruning (IMP), it is possible to extract a sparse subnetwork from a randomly initialized dense network that generalizes equally as well when trained (termed a 'winning' ticket). However, this technique is only effective for small datasets and networks such as a LeNet-300-100 (LeCun et al., 2002) trained on MNIST, and starts to break down for more complex settings (Liu et al., 2019; Frankle et al., 2020).

To remedy this, it was proposed to start from a slightly pretrained network, and extract a sparse network from that (termed a 'matching' ticket). Determining the required amount of pretraining to find such a ticket is often a difficult task requiring trial-and-error. One approach to do this was given in Frankle et al. (2020) which links stability to SGD noise with the existence of matching tickets. However, the authors do not give any theoretical explanation for this and do not show whether this is a necessary condition, or a sufficient condition.

Furthermore, as the IMP procedure involves repeatedly training networks to convergence, and then pruning with limited pruning rates, it is expensive to extract such a ticket, leading to several techniques being developed that limit the computational cost by reducing the number of training epochs (Zullich et al., 2021; You et al., 2020).

In this paper, we aim to find 'winning' tickets irrespective of the complexity of the considered setting. To accomplish this, we look at the impact different training hyperparameters have on the stability to SGD noise. We then compare the resulting tickets on various properties, including few-shot learning and transferability.

### 1.1 Contributions

- We consider different hyperparameter configurations for training dense networks and their impact on stability to SGD noise. While none of the hyperparameters lead to stability to SGD noise for a random initialization, applying IMP with some configurations nevertheless results in winning tickets without the need for a pretraining step, going against common knowledge. We term these hyperparameter configurations 'early-stable'.

- We demonstrate that, by limiting the training budget during IMP, the significant accuracy degradation that late-stable tickets experience at higher sparsities can be counteracted.

- We show that early-stable tickets generalize significantly better on limited datasets, even outperforming the dense network, leading us to believe this information is encoded in the sparsity mask. We show this is true by training linear classifiers to remarkable accuracy on the frozen tickets.

- We hypothesize, based on the training behaviour of different hyperparameters, that pruning a solution further from initialization negatively impacts the generalization of the resulting tickets. To reinforce this belief, we provide an experiment where we regularize this distance for late-stable hyperparameters, resulting in better performing tickets.

## 2 Background

### 2.1 (Linear) Mode Connectivity

To determine whether two networks are in the same loss basin – which is a connected region of low loss – the classification error over an interpolating path between their weights can be calculated. The error barrier is then defined as the maximal increase in classification error, and if this metric is sufficiently low, both networks lie within the same loss basin. This technique is called Linear Mode Connectivity (Frankle et al., 2020). A network is called stable to SGD noise if it always converges to the same loss basin when trained with SGD using different random seeds.

$$sup_\alpha \; \mathcal{E}(\alpha\theta_1 + (1-\alpha)\theta_2) - \frac{\mathcal{E}(\theta_1) + \mathcal{E}(\theta_2)}{2} \tag{1}$$

In practice, following the definition of Frankle et al. (2020), two networks are deemed to be in the same loss basin if the error barrier is lower than 2%. This threshold is also used to determine whether a network is stable to SGD noise.

**Other definitions of loss basins.** Linear Mode Connectivity is a restrictive way to determine whether two networks are connected in a low-loss region, and is not applicable to any two converged solutions. This however is the case when considering continuous paths. This has been shown by Garipov et al. (2018) using polynomial paths with a single bend or quadratic Bezier curves, or by an optimization based approach in (Draxler et al., 2018). Even then, with the additional trick of weight permutation, Entezari et al. (2022) posit that linear mode connectivity can be achieved for any two solutions. This proposition was then proven via Optimal Transport in (Ferbach et al., 2024).

Throughout this paper we use the definition of stability to SGD noise based on linear mode connectivity and construct the interpolating curve with 101 equally-spaced points i.e., $\alpha \in \{0, 0.01, ..., 0.99, 1.0\}$. For brevity, we shorten the term 'stable to SGD noise' in this paper to 'stable'.

---

**Algorithm 1** Iterative Magnitude Pruning

---

1: **Given:** A neural network initialized with weights $\theta_0 \in \mathbb{R}^d$, a pretraining budget of $p$ epochs, a training budget $T$, a number of pretraining iterations $N$, and a pruning rate $k$.
2: Initialize a binary pruning mask $M_0 = \mathbb{1}^d$.
3: Train $\theta_0$ for $p$ epochs to $\theta_{pt}$.      $\triangleright$ *Pretraining*
4: **for** level $L \in \{1, ..., N\}$ **do**      $\triangleright$ *Mask Search*
5:     Let $M_{L-1} \odot \theta_L$ be the result of training $M_{L-1} \odot \theta_{pt}$ for $T - p$ epochs.
6:     Initialize $M_L$ as $M_{L-1}$.
7:     Let $M_L[i] = 0$ for the $k\%$ lowest magnitude connections in $M_{L-1} \odot \theta_L$.
8: **end for**
9: Train the lottery ticket $M_N \odot \theta_{pt}$ to completion.      $\triangleright$ *Sparse Training*

---

## 2.2   Iterative Magnitude Pruning

The Lottery Ticket Hypothesis states that within a randomly initialized dense network with weights $\theta_0$, there exists a sparse subnetwork defined by a binary mask $M$, such that the ticket $M \odot \theta_0$ achieves similar validation accuracy as the dense network when trained under the same conditions. If this condition is satisfied, these tickets are then called 'winning' tickets.

The common approach to extract a Lottery Ticket is Iterative Magnitude Pruning (IMP) introduced by (Frankle & Carbin, 2019). This approach consists of two (or three) phases and starts with a randomly initialized dense neural network with weights $\theta_0$. Frankle et al. (2020) discovered that winning tickets can only be found in stable networks, and that in more complex settings randomly initialized weights are unstable. As such, they introduce an optional *Pretraining* phase (line 3) to the algorithm to remedy this. The sparsity mask $M$ itself is constructed in the *Mask Search* phase by iteratively training the network to completion (line 5) and pruning the lowest magnitude weights of the resulting network with a fixed pruning rate $k$ (line 7). Finally, in the *Sparse Training* phase the ticket is trained.

Another technique to find winning tickets was introduced in Renda et al. (2020) with Learning Rate Rewinding (LRR). The crucial difference between this technique and IMP with late rewinding is that after each pruning step, the learning rate schedule is rewound while the unpruned weights are not. A recent study (Gadhikar & Burkholz, 2024) has shown that this technique can generate better masks than IMP due to it being more robust to parameter signs at initialization. These signs are critical to the performance of a pruning mask as demonstrated by Zhou et al. (2019); Gadhikar et al. (2025). We however focus on IMP as it is more commonly employed.

**Modifications to the *Mask Search* phase.** Several modifications have been proposed to this basic algorithm. Rather than using a fixed pruning rate $k$, Paul et al. (2023) calculate an optimal pruning rate at each level, which allowed for higher pruning rates, thus reducing the numbers of iterations required to find a winning lottery ticket. In a similar manner, both Zullich et al. (2021); You et al. (2020) aimed to reduce computational cost by modifying the budget $T$, and determined that even with a significantly lower budget, winning tickets could still be found. Finally Maene et al. (2021) shows that for sufficiently large batch sizes in combination with gradual warmup, all networks become stable, and as such contain winning tickets without pretraining.

In this research we will first focus on applying different hyperparameters in the *Mask Search* phase and determining how this impacts the stability of a network, but also how these hyperparameters affect the existence of winning tickets in the network.

## 2.3   Learning Distance

Defined as the $l_2$ distance between initialization and the final converged weights (Equation (2)), this metric is typically used as a proxy for generalization. Papers such as (Li et al., 2020; Nagarajan & Kolter, 2019; He et al., 2022) link a lower learning distance to limiting the model capacity and as such reducing overfitting

on noisy labels. Conversely, Jiang et al. (2019) argues that there exists a negative correlation between generalization and learning distance.

$$\mathcal{D}(M_i \odot \theta_i) = ||M_i \odot \theta_{pt} - M_i \odot \theta_i||_2 \tag{2}$$

## 3 Experimental Setup

As training a neural network can involve a significant amount of hyperparameters which all can affect the generalization in different ways, we need to limit our search space. Specifically, we restrict our experiments to choosing different values for the batch size, momentum, and training budget, as each of those have clear indications in the literature of their impact on generalization.

### 3.1 Considered Hyperparameters

**Batch size ($b$).** In the case of large datasets, it is often infeasible to use the whole dataset during each update step of the network. As such, it is common to use minibatches with size $b$ of the dataset to update a network iteratively via Stochastic Gradient Descent. Intuitively, if the batch size is smaller, then the resulting gradient is more influenced by the subsampling of the dataset, as the batch composition can be dramatically different from the composition of the full dataset. However, it has been shown in Keskar et al. (2017) that smaller batch sizes positively impact the generalization of a neural network on unseen data.

**Momentum ($\mu$).** Introduced as a technique to speed up gradient descent, momentum allows gradient information to be carried over from previous batches in the weight update by employing a factor $\mu \in [0, 1]$ to weight the gradient information of the previous batch. As such, the updates function similar to a exponential moving average (see Equations (3) and (4)). The application of momentum has been demonstrated to lead to better generalizing networks, and it has been posited by Jelassi & Li (2022) that this is due to the resulting classifiers generalizing on small-margin samples, rather than memorizing those samples. The authors additionally argue that the impact of momentum is more significant with higher batch sizes.

$$v_{t+1} = \mu v_t - \epsilon \nabla f(\theta_t) \tag{3}$$

$$\theta_{t+1} = \theta_t + v_{t+1} \tag{4}$$

**Training Budget ($T$).** Training a network is a process with many sources of noise such as mini batch gradients, data augmentation and more. It is evident that this noise accumulates during repeated training epochs. As such, one method to limit SGD noise during training of a network is to simply limit the computational budget of the training phase, rather than tackling the sources of noise. More specifically, we will limit number of training epochs in the *Mask Search* phase. When not mentioned explicitly, we use a training budget of 200eps (or 90eps for ImageNet-100)

In the main paper, we highlight results using common values for each hyperparameter ($\mu \in \{0.0, 0.9\}$, $b \in \{100, 256\}$) but we demonstrate in Appendix A.5 that the observed trends hold for several less commonly used values.

### 3.2 Fixed Hyperparameters

The other hyperparameters used for training are fixed for each dataset network combination in order to not confound the impact of the studied hyperparameters. These hyperparameters are based on existing configurations.

**General hyperparameters.** We train each network with a learning rate of 0.1 (0.2 for TinyImageNet) which is cosine annealed during the training process and combine this with a weight decay of 1E-4. To improve generalization, we apply several augmentations to the training dataset, namely normalization based on dataset statistics, random cropping and horizontal flipping.

Table 1: Dataset, network combinations used in this paper.

| | | Dataset Statistics | | |
|---|---|---|---|---|
| Network | Dataset | # Classes | # Train samples | Image Size |
| ResNet-18 | CIFAR-10 | 10 | 50,000 | $32 \times 32$ |
| ResNet-18 VGG-16 | CIFAR100 | 100 | 50,000 | $32 \times 32$ |
| ResNet-34 Swin-S/16 | TinyImageNet | 200 | 100,000 | $64 \times 64$ |
| ResNet-50 | ImageNet-100 | 100 | 126,689 | $224 \times 224$ |

**IMP specific parameters.** Each iteration $k = 20\%$ of the unpruned parameters are pruned, meaning that after N iterations of pruning, $(100\% - k)^N$ of the connections are remaining. In general we use 25 pruning iterations, but for more computationally expensive settings we use less iterations (20 for TinyImageNet, 15 for ImageNet-100). When discussing pretraining, we pretrain for 2 epochs, which represents 1% of the total training budget (or 2.2% for ImageNet100) unless otherwise mentioned.

### 3.3  Datasets and Networks

The datasets and network combinations that we use are listed in Table 1 in roughly increasing complexity. For the analysis in the main body, we focus on ResNet-34 + TinyImageNet, however we show that our observations hold for the main contributions in the appendix.

## 4  On Training Hyperparameters, Stability, and Winning Tickets

We first determine the stability of dense networks when trained under different hyperparameter configurations, as (Frankle et al., 2020) argues this is necessary to find winning tickets. Next, we need to ascertain the impact of hyperparameters on ticket generalization in each of the IMP phases, to conclude whether a ticket is winning due to its specific mask, or due to the way it is trained. Finally, we can combine both aspects to conclude whether stability to SGD noise is required to find winning tickets. Additional results for other dataset, network combinations can be found Appendix A

### 4.1  Stability in Dense Networks

We start with a randomly initialized neural network and train it with a certain hyperparameter configuration to completion under different random seeds to achieve three different trained networks. These trained networks can then be used to calculate the error barrier for a given configuration by using pairwise interpolation between each of the sets of weights. We also repeat this experiment starting from pretrained weights, rather than randomly initialized weights.

**No configuration is stable at initialization.** By looking at the error barriers in Figure 1, we can notice that neither at initialization, nor when 1% pretrained, the weights can be considered stable under any hyperparameter configuration, as the error barrier is above the 2% threshold. We do however notice that with some hyperparameters, namely $\mu = 0.0$, the error barrier decreases significantly when using the pretrained weights, while for others it does not.

**Hyperparameters significantly influence emergence of stability.** Motivated by the significant differences in error barriers of pretrained networks given different hyperparameter configurations, we want to determine the least amount of pretraining required for which a network becomes stable. As calculating stability to SGD noise is an expensive process, requiring multiple full training runs for each initialization,

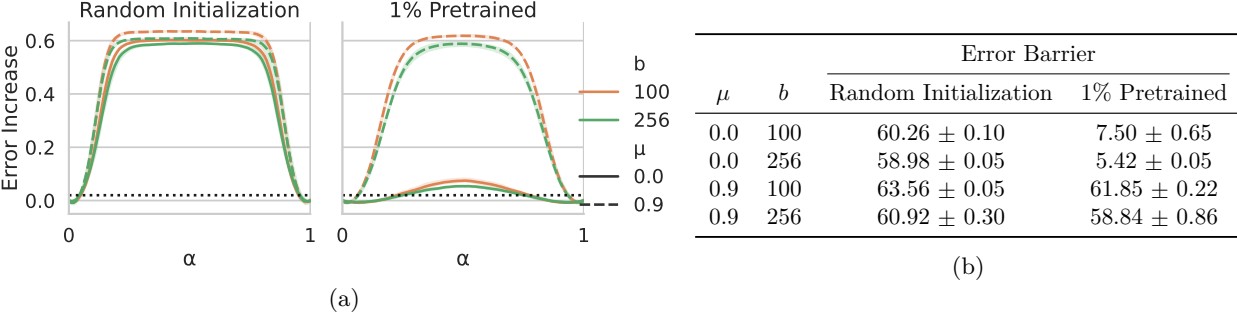

Figure 1: Instability to SGD noise for a dense ResNet-34 model trained on TinyImageNet with different training configurations. We show the full interpolation curve in **(a)** and the error barrier in **(b)**.

Table 2: The percentage of training elapsed at which point stability to SGD noise occurs for ResNet-34 on TinyImageNet trained under different hyperparameter settings.

|  | $b = 100$ | $b = 256$ |
|---|---|---|
| $\mu = 0.0$ | $2.50 \pm 0.0\%$ | $1.85 \pm 0.45\%$ |
| $\mu = 0.9$ | $18.35 \pm 0.95\%$ | $9.00 \pm 0.70\%$ |

we use a binary search algorithm[1] to speed up this process. This allows us to find the earliest stable rewind point in $\lfloor log_2 200 + 1 \rfloor = 8$ tries ($\lfloor log_2 90 + 1 \rfloor = 7$ for ImageNet100).

In Table 2 we notice that the first emergence of stability to SGD noise can vary significantly, ranging from occurring after 2% of the total training duration up to 18%. A later stability is associated with the application of momentum, or decreasing the batch size. In particular, the use of momentum ($\mu = 0.9$) delays stability most. These results follow our observations found in Figure 1.

## 4.2 Disentangling Mask Search and Sparse Training

It is evident that the final validation accuracy of a lottery ticket depends on all three phases of the IMP algorithm (see Algorithm 1). While the original approach of (Frankle & Carbin, 2019) uses the same hyperparameters throughout the algorithm, it is possible to select different hyperparameters for each phase. For simplicity, we focus our analysis on tickets without pretraining. To determine the impact of hyperparameter selection in these phases on the final generalization, we independently vary their values in this experiment.

**Generalization is primarily determined by Mask Search hyperparameters.** We can see in Table 3 that – for each combination of $\mu_{ST}, b_{ST}$ – tickets found with $\mu_{MS} = 0.0, b_{MS} = 256$ outperform those found with $\mu_{MS} = 0.9, b_{MS} = 100$ when each is trained with the same $\mu_{ST}, b_{ST}$. This leads us to believe that the primary driver behind the generalization of a lottery ticket is the structure of the sparsity mask, which is determined by the hyperparameters used in the Mask Search phase.

**Sparse Training hyperparameters refine generalization.** Even though the Mask Search hyperparameters most significantly impact validation accuracy, there is also a smaller role for the Sparse Training Hyperparameters which can be used to refine the performance of a ticket. Here we see that hyperparameters which improved dense network training, aid ticket generalization as well.

As we have determined that Mask Search hyperparameters are most significant for ticket generalization, we will follow existing convention throughout the rest of the paper, and use the same value for $\mu$ and $b$ in each of the phases of the IMP algorithm.

---

[1]This implies that we assume the error barrier decreases monotonically throughout training.

Table 3: Validation accuracy for ResNet-34 Tickets on TinyImageNet for different combinations of Mask Search ($\mu_{\mathrm{MS}}$, $b_{\mathrm{MS}}$) and Sparse Training ($\mu_{\mathrm{ST}}$, $b_{\mathrm{ST}}$) hyperparameters. Starred (*) entries encompass one or multiple runs in which the ticket did not converge and remained at random chance.

| Hyperparameters | | | | | Sparsity | | | | |
|---|---|---|---|---|---|---|---|---|---|
| $\mu_{\mathrm{MS}}$ | $b_{\mathrm{MS}}$ | Pretrained | $\mu_{\mathrm{ST}}$ | $b_{\mathrm{ST}}$ | 0.00% | 67.23% | 89.26% | 96.48% | 98.85% |
| 0.0 | 256 | ✗ | 0.0 | 100 | 61.14% | 60.18% | 60.60% | 58.98% | 55.94% |
| | | | 0.0 | 256 | 59.33% | 59.95% | 60.20% | 58.89% | **57.09%** |
| | | | 0.9 | 100 | **63.74%** | **62.83%** | **61.56%** | **59.48%** | 55.27% |
| | | | 0.9 | 256 | 61.57% | 60.78% | 59.49% | 57.27% | 54.45% |
| 0.9 | 100 | ✗ | 0.0 | 100 | 61.14% | 58.48% | 51.55% | 48.85% | 42.70% |
| | | | 0.0 | 256 | 59.33% | 56.73% | 49.74% | 47.09% | 41.34% |
| | | | 0.9 | 100 | **63.74%** | **62.46%** | 35.95%* | 35.69%* | 15.09%* |
| | | | 0.9 | 256 | 61.57% | 59.28% | **52.33%** | **49.58%** | **44.32%** |

## 4.3 Winning tickets without stability

The previous experiment did not yield any hyperparameters for which dense networks are stable at initialization, which leads us to expect that we can not find any winning tickets for these settings.

**Winning tickets can be found in unstable initializations.** We notice that winning tickets (underlined entries in Table 4) exist in random initializations when using the hyperparameter configuration $\{\mu = 0.0, b = 256\}$. While this configuration has been deemed unstable to SGD noise (see Figure 1), it is the configuration that becomes stable after the least amount of pretraining (see Table 2). This shows us that stability to SGD noise is not a hard prerequisite to find winning tickets, but rather this observation hints at a relationship between the point at which stability emerges and winning tickets. As such, we will refer to such hyperparameters as *early-stable* hyperparameters.

Table 4: Validation accuracy of Lottery Ticket extracted at different sparsity levels from ResNet-34 on TinyImageNet given different hyperparameters. Winning tickets are underlined. Starred(*) entries encompass one or multiple runs in which the ticket did not converge and remained at random chance.

| Hyperparameters | | | Sparsity | | | | |
|---|---|---|---|---|---|---|---|
| $\mu$ | $b$ | Pretrained | 0.00% | 67.23% | 89.26% | 96.48% | 98.85% |
| 0.0 | 100 | ✗ | 61.14% | 59.76% | 57.79% | 54.47% | 50.98% |
| 0.0 | 256 | | 59.33% | 59.95% | **60.20%** | **58.89%** | **57.09%** |
| 0.9 | 100 | | **63.74%** | 62.46% | 35.95%* | 35.69%* | 15.09%* |
| 0.9 | 256 | | 61.57% | 60.37% | 56.93% | 52.76% | 47.34% |
| 0.0 | 100 | ✓ | 60.91% | 62.90% | **62.67%** | **60.37%** | 56.11% |
| 0.0 | 256 | | 59.27% | 60.01% | 60.08% | 58.59% | **56.40%** |
| 0.9 | 100 | | **63.45%** | **63.02%** | 60.10% | 57.73% | 35.10%* |
| 0.9 | 256 | | 61.66% | 61.22% | 58.95% | 55.91% | 52.49% |

**Stability can emerge after several *Mask Search* iterations.** Paul et al. (2023) show that if dense networks are stable to SGD noise, then the derived tickets are also stable, and tickets of level $L$ and $L + 1$ can be linearly connected with a low error barrier. While have demonstrated that winning tickets can be found in unstable dense networks, we are still interested if this lack of stability persists when sparsifying the network. To this end, we plot the linear interpolation curves between lottery tickets at different sparsities in Figure 2. We can clearly see that for the earliest-stable configuration, tickets become fully stable to SGD noise after several rounds of pruning, which is not the case for other hyperparameter configurations.

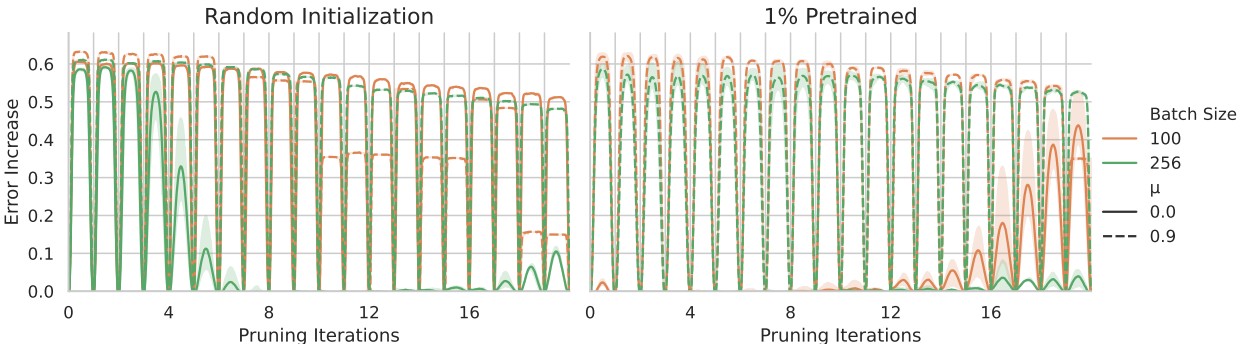

Figure 2: Error increase when linearly interpolating between subsequent ResNet-34 + TinyImageNet tickets.

### 4.4 How budget impacts ticket generalization

SGD noise accumulates throughout training, and as such one can expect that limiting the training budget limits the impact of improves the stability to SGD noise of the network. In addition, this would have the beneficial effect that it also limits the computational cost associated with IMP. From this second angle, existing methodologies such as Accelerated Iterative Magnitude Pruning (AIMP) (Zullich et al., 2021) have already been introduced, and have observed that the training budget in the *Mask Search* phase can be limited without significant impact on the resulting ticket performance. We employ a budget of either 20 or 50 epochs which represents 10% and 25%, respectively, of the original budget. The LR schedule is modified to reflect upon this limited budget. As we study the impact on the sparsity mask, the *Sparse Training* phase is kept unchanged and uses the full budget.

Table 5: Accelerated Iterative Magnitude Pruning (AIMP) results for different hyperparameter settings of ResNet-34 on TinyImageNet. Starred (*) entries encompass one or multiple runs in which the ticket did not converge and remained at random chance.

| Hyperparameters | | | Sparsity | | | | |
|---|---|---|---|---|---|---|---|
| Budget | $\mu$ | $b$ | 0.00% | 67.23% | 89.26% | 96.48% | 98.85% |
| | 0.0 | 100 | 61.14% | 60.78% | **61.07%** | **60.10%** | **58.61%** |
| | 0.0 | 256 | 59.33% | 59.88% | 60.09% | 59.01% | 57.08% |
| 20eps | 0.9 | 100 | **63.74%** | **63.20%** | 61.05% | 58.58% | 53.52% |
| | 0.9 | 256 | 61.57% | 61.45% | 59.94% | 57.30% | 53.45% |
| | 0.0 | 100 | 61.14% | **61.10%** | **61.27%** | **60.83%** | **58.63%** |
| | 0.0 | 256 | 59.33% | 59.79% | 60.12% | 59.40% | 56.65% |
| 50eps | 0.9 | 100 | **63.74%** | 62.82% | 59.48% | 53.64% | 38.93% |
| | 0.9 | 256 | 61.57% | 60.42% | 58.76% | 55.92% | 53.55% |

**Late-stable hyperparameters benefit from a limited budget.** We can see in Table 5 that limiting the budget results in similar or better performing lottery tickets for all considered configurations of $\mu$, $b$. While in the case of the early-stable hyperparameters, the validation accuracy of tickets found with a low budget is similar to that of the tickets found with the full budget (see Table 4), in the other cases the validation accuracy increases when decreasing the budget. This indicates that these masks – found early during training – are actually better trainable than those that emerge when the network has converged.

## 5 The Emergence of Generalization in Early-stable Tickets

Our next objective is to determine what makes early-stable tickets winning. For this we want to determine whether these tickets have an inherent advantage leading to better generalization that the other non-winning

Table 6: Validation accuracies of 89.26% sparse ResNet-34 tickets when trained on a TinyImageNet subset. Entries annotated with a † have a sparsity of 0.00%.

| Hyperparameters | | | Subsampling rate | | | |
|---|---|---|---|---|---|---|
| $\mu$ | $b$ | Budget | 1% | 2% | 5% | 10% |
| 0.0 | 100 | | $20.70 \pm 0.41\%$ | $27.06 \pm 0.40\%$ | $37.09 \pm 0.20\%$ | $44.13 \pm 0.54\%$ |
| 0.0 | 256 | 20eps | $\mathbf{28.49 \pm 0.53\%}$ | $\mathbf{33.06 \pm 0.68\%}$ | $\mathbf{39.69 \pm 0.20\%}$ | $\mathbf{46.18 \pm 0.54\%}$ |
| 0.9 | 100 | | $3.89 \pm 0.19\%$ | $6.77 \pm 0.25\%$ | $13.56 \pm 0.68\%$ | $24.74 \pm 1.09\%$ |
| 0.9 | 256 | | $4.75 \pm 0.38\%$ | $8.02 \pm 0.63\%$ | $16.20 \pm 0.66\%$ | $24.81 \pm 1.15\%$ |
| 0.0 | 100 | | $4.72 \pm 0.46\%$ | $7.69 \pm 0.39\%$ | $16.46 \pm 0.24\%$ | $27.34 \pm 0.18\%$ |
| 0.0 | 256 | 200eps | $\mathbf{10.49 \pm 1.14\%}$ | $\mathbf{17.90 \pm 1.72\%}$ | $\mathbf{29.51 \pm 1.71\%}$ | $\mathbf{37.87 \pm 0.43\%}$ |
| 0.9 | 100 | | $3.41 \pm 0.28\%$ | $5.18 \pm 0.59\%$ | $14.33 \pm 0.55\%$ | $8.36 \pm 13.61\%$ |
| 0.9 | 256 | | $3.40 \pm 0.62\%$ | $6.17 \pm 0.85\%$ | $11.42 \pm 0.31\%$ | $21.25 \pm 1.63\%$ |
| Permuted Ticket | | | $4.34 \pm 0.20\%$ | $6.33 \pm 0.48\%$ | $13.75 \pm 0.43\%$ | $26.30 \pm 0.31\%$ |
| Dense Network † | | | $4.32 \pm 0.13\%$ | $6.33 \pm 0.19\%$ | $13.19 \pm 0.33\%$ | $25.13 \pm 0.57\%$ |

tickets do not have. We do this by first limiting the data availability in the *Sparse Training* phase. Afterwards we directly study the predictive power of untrained Lottery tickets, and how transferable the untrained features are to other datasets. Additional results for other dataset, network combinations can be found Appendix B.

## 5.1 Few-shot generalizability

By limiting the number of data samples in the *Sparse Training* phase, we can see how well tickets found with different hyperparameter settings generalize when less data is available. In our experiments we subsample the dataset via the random selection method – as this has been shown to work best for small subsets (Guo et al., 2022) while being also computationally efficient to calculate – using the following rate $[1\%, 2\%, 5\%, 10\%]$, while making sure that the resulting subset is class-balanced. We additionally compare with two baselines, namely a fully dense network, and a network obtained by layerwise permutation of the sparsity mask of a Lottery Ticket, thus having the same layerwise sparsity and initialization. Results for the 89.26% sparsity are listed in Table 6.

**Early-stable tickets generalize better with limited data.** We observe that early-stable tickets can recover a significantly higher proportion of the full dataset accuracy when trained on a dataset subset. More specifically, they can recover $\sim 65\%$ of the full dataset accuracy given 10 times less data, where other networks recover less than 50%. Limiting the training budget of the *Mask Search* phase to 20eps results in an even more impressive recovery of $\sim 78\%$ for the early-stable tickets, while having a negligible impact for the other hyperparameters.

## 5.2 Features encoded in tickets

We have seen that early-stable tickets generalize considerably better than other networks with limited data in the *Sparse Training* phase. Next we employ a technique which foregoes the *Sparse Training* phase completely. Earlier work by Zhou et al. (2019) proved the existence of supermasks, which are tickets that can achieve better than random chance predictions without any training. While these have been found on very simple settings with a different pruning criterion, we were unable to replicate these results on more complex settings. Instead we consider a less powerful phenomenon. Rather than keeping both feature extractor and classification layer fixed at initialization, instead we only keep the feature extractor fixed, and train a linear layer on top of the features in the network, following the approach of (Alain & Bengio, 2017). We compare with the same baselines as in the previous experiment.

Table 7: Validation accuracy for a linear classifier trained on *frozen* ResNet-34 lottery tickets at different sparsities. Entries annotated with a † have a sparsity of 0.00%.

| Hyperparameters | | | Sparsities | | | |
|---|---|---|---|---|---|---|
| $\mu$ | $b$ | Budget | 67.23% | 89.26% | 96.48% | 98.85% |
| 0.0 | 100 | | $7.51 \pm 0.49\%$ | $9.67 \pm 0.23\%$ | $9.58 \pm 0.22\%$ | $8.28 \pm 0.42\%$ |
| 0.0 | 256 | 20eps | $\mathbf{8.87 \pm 0.42\%}$ | $\mathbf{10.11 \pm 0.15\%}$ | $\mathbf{10.55 \pm 0.25\%}$ | $\mathbf{9.75 \pm 0.75\%}$ |
| 0.9 | 100 | | $3.89 \pm 0.10\%$ | $3.60 \pm 0.24\%$ | $3.42 \pm 0.19\%$ | $3.46 \pm 0.48\%$ |
| 0.9 | 256 | | $3.84 \pm 0.28\%$ | $4.34 \pm 0.34\%$ | $4.06 \pm 0.31\%$ | $3.82 \pm 0.14\%$ |
| 0.0 | 100 | | $3.83 \pm 0.19\%$ | $3.84 \pm 0.27\%$ | $3.85 \pm 0.37\%$ | $3.36 \pm 0.12\%$ |
| 0.0 | 256 | 200eps | $\mathbf{4.53 \pm 0.47\%}$ | $\mathbf{5.53 \pm 0.70\%}$ | $\mathbf{5.02 \pm 0.32\%}$ | $\mathbf{4.29 \pm 0.09\%}$ |
| 0.9 | 100 | | $1.89 \pm 0.27\%$ | $1.59 \pm 0.40\%$ | $1.76 \pm 0.50\%$ | $1.65 \pm 0.21\%$ |
| 0.9 | 256 | | $2.63 \pm 0.53\%$ | $2.04 \pm 0.90\%$ | $1.93 \pm 0.44\%$ | $1.97 \pm 0.37\%$ |
| Permuted Ticket | | | $3.39 \pm 0.09\%$ | $4.09 \pm 0.15\%$ | $0.50 \pm 0.00\%$ | $0.50 \pm 0.00\%$ |
| Dense Network † | | | | $3.13 \pm 0.45\%$ | | |

**Early-stable tickets encode useful features.** From the resulting validation accuracies listed in Table 7 we see that in early-stable tickets the encoded features are better than both baselines. Sparsity of the network has a small part in this property, as we see that validation accuracy for most configurations are increasing with sparsity, up to 89.26% sparsity, after which they drop slightly. We can see significant parallels between these results and those in Table 6 where a worse generalization of the frozen features corresponds to a worse few-shot generalizability.

## 5.3 Transferability

Finally, we ask ourselves whether those features that emerge in the network by pruning are uniquely optimized for the dataset the network is trained on. To evaluate this, we keep the feature extractor of the network frozen, as in the previous experiment, and train a new (dense) linear layer on the final features with the output classes adapted to the new dataset. This incurs a slight loss of sparsity, but is preferable over resparsifying the new classification layer due to computational cost. For transferring, we follow the regime of the *Sparse Training* phase. We transfer to the following datasets, roughly in order of increasing difficulty : EuroSAT (Helber et al., 2019), CIFAR-10 (Krizhevsky et al., 2009), CIFAR-100 (Krizhevsky et al., 2009), CUB-200 (Wah et al., 2011). These differ from TinyImageNet by either having different types of classes (EuroSAT), different complexities (CIFAR-10, CIFAR-100), or higher resolution and more fine-grained classes (CUB-200). We compare the results with transferring a frozen permuted ticket, and a frozen *trained* dense network.

**Sparse features can outperform dense *trained* features.** For all datasets except CUB-200 we can observe that the features in an early-stable ticket at initialization transfer better than the features in a *trained* dense network. Specifically this is the case in settings with a limited *Mask Search budget*. When we use the full budget, we can still see remarkable performance, but the features in those tickets underperform slightly. As expected, the features that perform best on the original dataset also transfer best to another dataset.

## 6 Finding Tickets near the Initialization

A common thread throughout our experiments is that hyperparameters which positively impact generalization – such as using a smaller batch size, employing momentum ($\mu = 0.9$) – negatively impact the stability to SGD noise (see Table 2). This negative impact on stability translates in worse pruning efficiency and underperforming sparse structures in the tickets, which is made clear in the disentanglement experiments (see Table 3). We hypothesize that the cause of this instability lies in the specific optimization behavior of networks trained under these hyperparameters, and more specifically justify this using the learning distance metric.

Table 8: Transferability to different datasets for frozen 89.26% ResNet-34 tickets extracted on TinyImageNet with different configurations. Entries annotated with a † have a sparsity of 0.00%.

| Hyperparameters | | | Target dataset | | | |
|---|---|---|---|---|---|---|
| $\mu$ | $b$ | Budget | EuroSAT | CIFAR-10 | CIFAR-100 | CUB-200 |
| 0.0 | 100 | | $92.04 \pm 0.42\%$ | $\mathbf{68.11 \pm 0.69\%}$ | $45.63 \pm 0.76\%$ | $14.08 \pm 0.17\%$ |
| 0.0 | 256 | | $\mathbf{92.37 \pm 0.26\%}$ | $67.84 \pm 0.59\%$ | $\mathbf{45.95 \pm 0.56\%}$ | $\mathbf{14.78 \pm 0.39\%}$ |
| 0.9 | 100 | 20eps | $65.75 \pm 0.63\%$ | $30.21 \pm 0.52\%$ | $45.63 \pm 0.76\%$ | $3.21 \pm 0.23\%$ |
| 0.9 | 256 | | $73.45 \pm 0.74\%$ | $40.17 \pm 1.25\%$ | $17.53 \pm 0.50\%$ | $4.14 \pm 0.14\%$ |
| 0.0 | 100 | | $63.25 \pm 0.76\%$ | $27.87 \pm 0.60\%$ | $10.25 \pm 0.63\%$ | $3.57 \pm 0.27\%$ |
| 0.0 | 256 | | $\mathbf{78.19 \pm 0.30\%}$ | $\mathbf{46.72 \pm 1.98\%}$ | $\mathbf{22.86 \pm 1.06\%}$ | $\mathbf{5.74 \pm 0.30\%}$ |
| 0.9 | 100 | 200eps | $46.19 \pm 8.49\%$ | $26.54 \pm 1.83\%$ | $6.98 \pm 1.12\%$ | $3.24 \pm 0.16\%$ |
| 0.9 | 256 | | $54.25 \pm 5.52\%$ | $26.83 \pm 2.69\%$ | $8.87 \pm 1.71\%$ | $3.92 \pm 1.34\%$ |
| Permuted Ticket | | | $65.03 \pm 4.09\%$ | $30.72 \pm 0.60\%$ | $12.13 \pm 0.72\%$ | $3.89 \pm 0.35\%$ |
| *Trained* Dense Network † | | | $90.28 \pm 0.52\%$ | $65.11 \pm 0.19\%$ | $44.24 \pm 0.02\%$ | $20.92\% \pm 0.57\%$ |

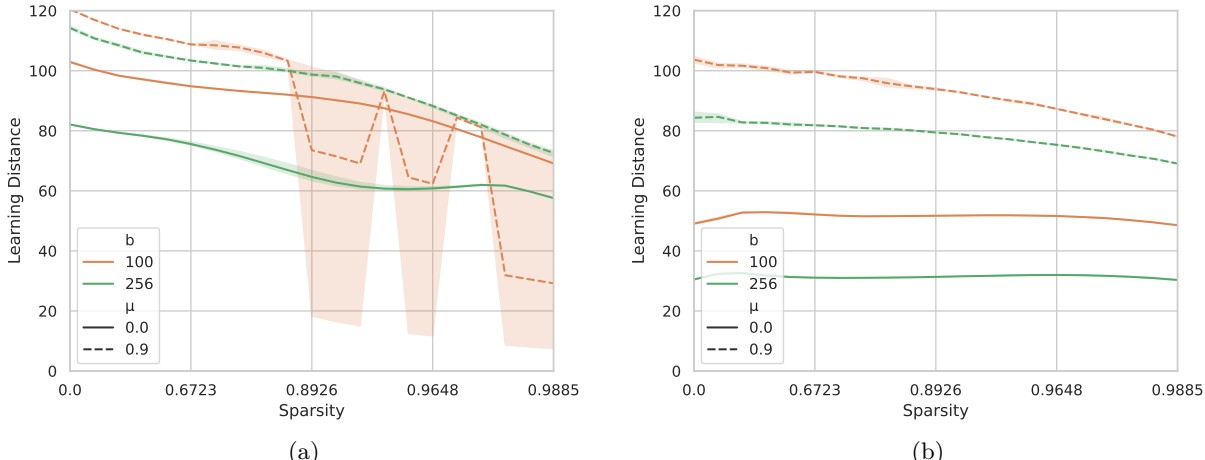

(a)      (b)

Figure 3: Learning distance in the *Mask Search* phase given different hyperparameter settings for ResNet-34 + TinyImageNet with both **(a)** full budget, and **(b)** limited budget. Notice that at several sparsities the learning distance is significantly lower due to some runs not converging.

**Early-stable tickets converge closer to the initialization.** When calculating this metric, we notice that a lower distance from initialization aligns well with the early-stable hyperparameters we determined to produce the best pruning mask (Figure 3). We hypothesize that this is precisely why additional pretraining is required to find winning tickets in certain settings (e.g. $\{\mu = 0.9, b = 100\}$), as those settings have a higher learning distance at initialization and pretraining can limit learning distance. By converging to a solution further from the initialization, a disconnect can emerge between low-magnitude connections at convergence, and low-magnitude connections in early training close to initialization. This means that when applying iterative pruning, connections which could be valuable during the optimization process can be removed, as they have a low magnitude at convergence. This inefficiency can accumulate and result in significant accuracy drops at high sparsity.

**Regularizing the learning distance.** To verify the impact of learning distance on the IMP procedure, we base ourselves on a suggestion made in (Li et al., 2020) by adding an additional term to the objective which minimizes the learning distance in the *Mask Search* phase, balanced by a factor $\tau$. Compared to using different hyperparameters, this approach limits the learning distance artificially, but it is effective

|  | Sparsity | | | | |
|---|---|---|---|---|---|
|  | 0.00% | 67.23% | 89.26% | 96.48% | 98.85% |
| $\tau = 0.0$ | 63.74% | 62.46% | 35.95%* | 35.69%* | 15.09%* |
| $\tau = 0.001$ | 63.74% | 61.98% | 40.50%* | 51.96% | 25.04%* |
| $\tau = 0.005$ | 63.74% | 62.67% | 60.62% | 57.14% | 51.67% |
| $\tau = 0.01$ | 63.74% | 62.93% | 61.06% | 58.35% | **53.37%** |
| $\tau = 0.05$ | 63.74% | **63.52%** | **62.63%** | **60.93%** | 51.36% |

(a)

(b)

Figure 4: **(a)** Validation accuracy, **(b)** Learning distance for tickets found with ResNet-34 on TinyImageNet ($\{\mu = 0.9,\ b = 100\}$) without pretraining when regularized. Starred(*) entries encompass one or multiple runs in which the ticket did not converge and remained at random chance.

(see Figure 4b). Employing this technique leads to significant increases in generalization accuracy at higher sparsities depending on the regularization strength (see Figure 4a), reinforcing the belief that pruning a solution closer to the initialization limits the impact on generalization w.r.t. the dense network.

## 7 Conclusion

In this paper, we analyzed the impact of hyperparameter selection on the stability to SGD noise of a dense network which is used in the literature as a prerequisite for finding winning lottery tickets. We found that when using certain hyperparameter configurations stability, the dense network is unstable to SGD noise, but winning tickets can be extracted, thus negating the need for pretraining. We termed these hyperparameters and their resulting tickets early-stable, as they were unstable at initialization, but required limited pretraining to become stable. By limiting the *Mask Search* budget we could counteract some of the accuracy loss of the late-stable hyperparameters.

These early-stable hyperparameters have additional valuable properties outside of the existence of winning tickets. We discovered that their resulting tickets generalize significantly better with limited data than even a dense network. This lead us to determine that within such tickets better generalizable features exist by virtue of the pruning mask. We showed that these features significantly outperform random dense features, and transfer better to other datasets than the features in a *trained* dense network in certain cases.

To elucidate these phenomena, we turn to an explanation based on the learning distance metric. We show that training with early-stable hyperparameters result in a solution that is significantly closer to the initialization than training with other hyperparameters. Next we employed a regularization minimizing the learning distance together with late-stable hyperparameters and determined that this significantly reduced the accuracy degradation when pruning. This leads us to hypothesize a connection between the learning distance and the existence of winning tickets.

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

## Appendices

- Appendix A contains additional results for Section 4 on different dataset-network combinations. More specifically, we show how stability emerges in dense networks, the impact of hyperparameter selection in different phases, that winning tickets do exist for settings without stability, and how *Mask Search* budget impacts ticket generalization. Additionally, we show the impact of more extreme hyperparameter values on ResNet-18 + CIFAR-10, and explore other hyperparameters on ResNet-34 + TinyImageNet.

- In Appendix B we extend the observations of Section 5. This means highlighting the few-shot generalizability of early-stable tickets, the features that are encoded in those tickets, and their transferability to other datasets.

- In Appendix C we show the impact of hyperparameters on learning distance for different dataset-network combinations, including the additional hyperparameters studied in Appendix A. We furthermore show that regularizing the learning distance is applicable to a variety of settings.

- Hyperparameter configurations for all experiments conducted in this work are listed in Appendix D.

## A  On Training Hyperparameters, Stability, and Winning Tickets

### A.1  Stability in Dense Networks

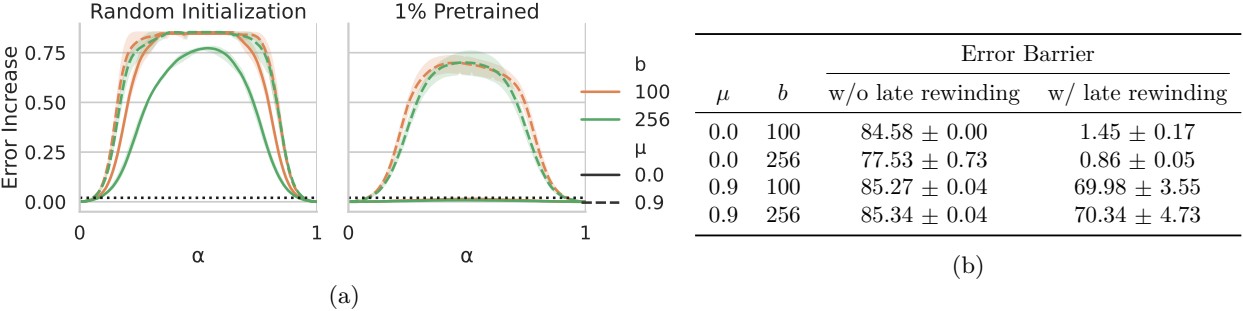

| | | Error Barrier | |
|---|---|---|---|
| $\mu$ | $b$ | w/o late rewinding | w/ late rewinding |
| 0.0 | 100 | $84.58 \pm 0.00$ | $1.45 \pm 0.17$ |
| 0.0 | 256 | $77.53 \pm 0.73$ | $0.86 \pm 0.05$ |
| 0.9 | 100 | $85.27 \pm 0.04$ | $69.98 \pm 3.55$ |
| 0.9 | 256 | $85.34 \pm 0.04$ | $70.34 \pm 4.73$ |

(a)        (b)

Figure 5: Instability for a dense **ResNet-18** model trained on **CIFAR10** with different training configurations. We show the full interpolation curve in **(a)** and the error barrier in **(b)**.

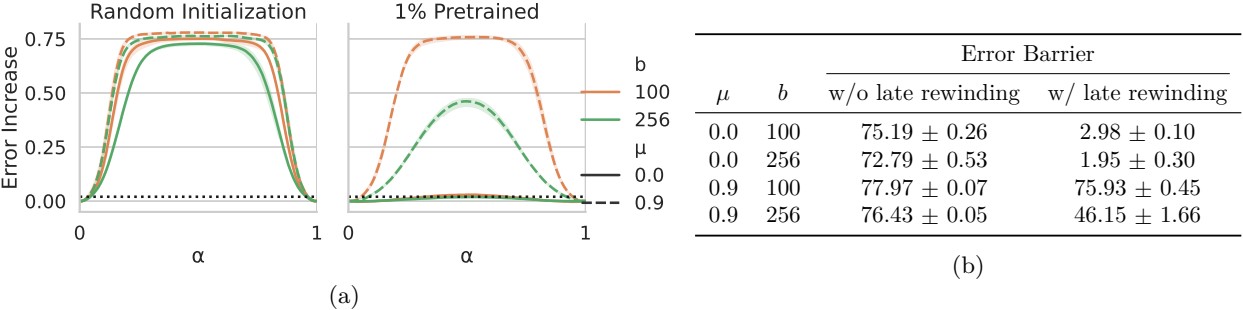

| | | Error Barrier | |
|---|---|---|---|
| $\mu$ | $b$ | w/o late rewinding | w/ late rewinding |
| 0.0 | 100 | $75.19 \pm 0.26$ | $2.98 \pm 0.10$ |
| 0.0 | 256 | $72.79 \pm 0.53$ | $1.95 \pm 0.30$ |
| 0.9 | 100 | $77.97 \pm 0.07$ | $75.93 \pm 0.45$ |
| 0.9 | 256 | $76.43 \pm 0.05$ | $46.15 \pm 1.66$ |

(a)        (b)

Figure 6: Instability for a dense **ResNet-18** model trained on **CIFAR100** with different training configurations. We show the full interpolation curve in **(a)** and the error barrier in **(b)**.

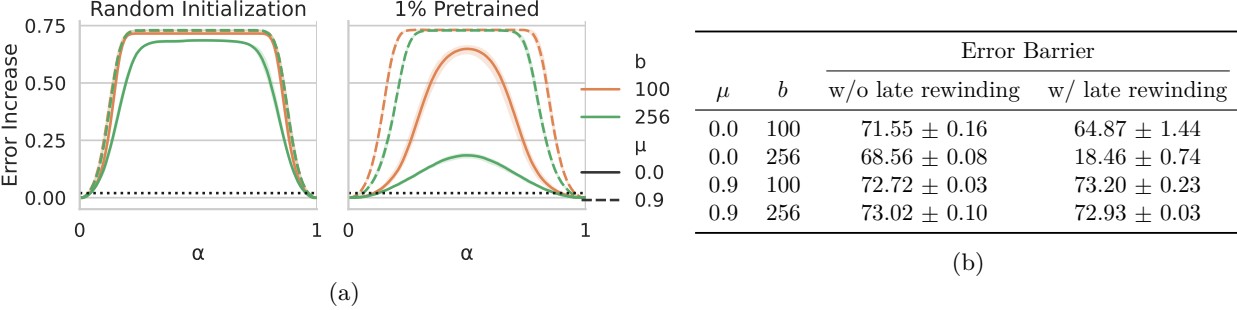

Figure 7: Instability for a dense **VGG16** model trained on **CIFAR100** with different training configurations. We show the full interpolation curve in **(a)** and the error barrier in **(b)**.

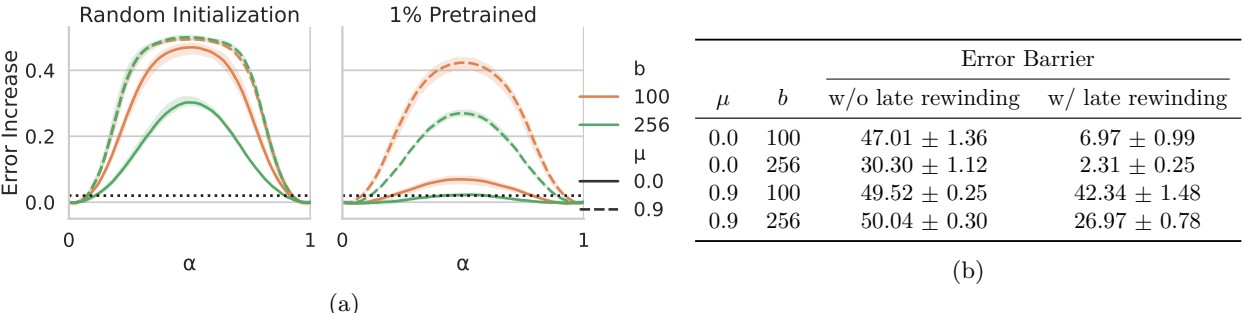

Figure 8: Instability for a dense **Swin** model trained on **TinyImageNet** with different training configurations. We show the full interpolation curve in **(a)** and the error barrier in **(b)**.

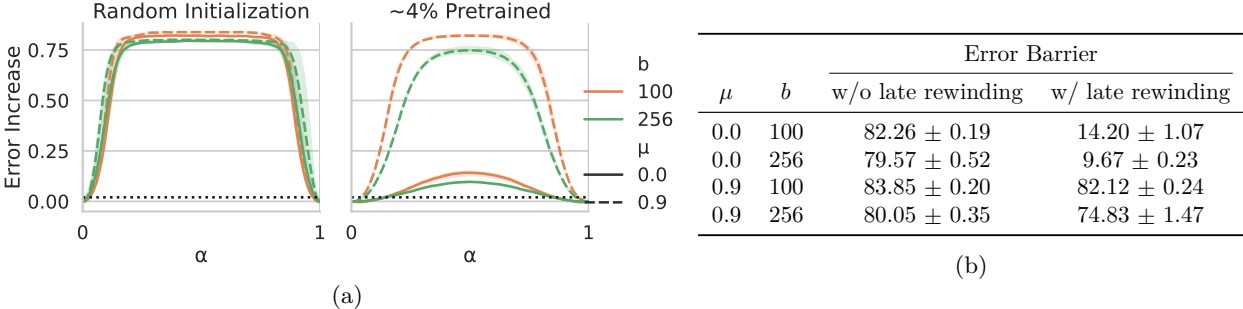

Figure 9: Instability for a dense **ResNet-50** model trained on **ImageNet-100** with different training configurations. We show the full interpolation curve in **(a)** and the error barrier in **(b)**.

Table 9: The percentage of training elapsed at which point stability to SGD noise occurs for the considered model and dataset combinations trained under different hyperparameter settings.

| | | $\mu = 0.0$ | | $\mu = 0.9$ | |
|---|---|---|---|---|---|
| Network | Dataset | $b = 100$ | $b = 256$ | $b = 100$ | $b = 256$ |
| ResNet-18 | CIFAR-10 | $1.15 \pm 0.25\%$ | $\mathbf{0.85 \pm 0.25\%}$ | $21.35 \pm 0.95\%$ | $11.85 \pm 0.60\%$ |
| ResNet-18 | CIFAR-100 | $1.50 \pm 0.00\%$ | $\mathbf{1.00 \pm 0.00\%}$ | $23.65 \pm 1.05\%$ | $9.50 \pm 0.40\%$ |
| VGG16 | CIFAR-100 | $6.65 \pm 0.25\%$ | $\mathbf{4.15 \pm 0.25\%}$ | $42.50 \pm 0.40\%$ | $20.15 \pm 0.25\%$ |
| Swin | TinyImageNet | $2.50 \pm 0.00\%$ | $\mathbf{1.00 \pm 0.00\%}$ | $19.15 \pm 0.29\%$ | $8.35 \pm 0.75\%$ |
| ResNet-50 | ImageNet-100 | $6.67 \pm 0.00\%$ | $\mathbf{6.33 \pm 0.56\%}$ | $28.56 \pm 2.33\%$ | $18.56 \pm 2.33\%$ |

## A.2 Disentangling Mask Search and Sparse Training

Table 10: Validation accuracy for **ResNet-18** Tickets on **CIFAR-10** for different combinations of Mask Search ($\mu_{\text{MS}}$, $b_{\text{MS}}$) and Sparse Training ($\mu_{\text{ST}}$, $b_{\text{ST}}$) hyperparameters.

| Hyperparameters | | | | | Sparsity | | | | |
|---|---|---|---|---|---|---|---|---|---|
| $\mu_{\text{MS}}$ | $b_{\text{MS}}$ | Pretrained | $\mu_{\text{ST}}$ | $b_{\text{ST}}$ | 0.00% | 67.23% | 89.26% | 96.48% | 98.85% |
| 0.0 | 256 | ✗ | 0.0 | 100 | 94.58% | 94.55% | 94.37% | **93.79%** | **92.80%** |
| | | | 0.0 | 256 | 93.63% | 94.02% | 93.93% | 93.39% | 92.67% |
| | | | 0.9 | 100 | 95.24% | **95.11%** | **94.78%** | 93.68% | 91.82% |
| | | | 0.9 | 256 | 95.18% | 94.99% | 94.56% | 93.55% | 92.05% |
| 0.9 | 100 | ✗ | 0.0 | 100 | 94.58% | 94.05% | 93.14% | 91.99% | 90.06% |
| | | | 0.0 | 256 | 93.63% | 92.85% | 92.30% | 90.79% | 89.07% |
| | | | 0.9 | 100 | 95.24% | **94.99%** | **94.39%** | **93.25%** | **91.41%** |
| | | | 0.9 | 256 | 95.18% | 94.72% | 94.13% | 93.16% | 91.22% |

Table 11: Validation accuracy for **VGG16** Tickets on **CIFAR-100** for different combinations of Mask Search ($\mu_{\text{MS}}$, $b_{\text{MS}}$) and Sparse Training ($\mu_{\text{ST}}$, $b_{\text{ST}}$) hyperparameters.

| Hyperparameters | | | | | Sparsity | | | | |
|---|---|---|---|---|---|---|---|---|---|
| $\mu_{\text{MS}}$ | $b_{\text{MS}}$ | Pretrained | $\mu_{\text{ST}}$ | $b_{\text{ST}}$ | 0.00% | 67.23% | 89.26% | 96.48% | 98.85% |
| 0.0 | 256 | ✗ | 0.0 | 100 | 72.77% | 70.27% | 69.11% | 67.74% | **65.98%** |
| | | | 0.0 | 256 | 69.26% | 69.53% | 68.80% | 67.54% | 65.41% |
| | | | 0.9 | 100 | 73.90% | **73.30%** | **71.73%** | **69.29%** | 62.79% |
| | | | 0.9 | 256 | **73.98%** | 72.57% | 70.19% | 68.60% | 65.43% |
| 0.9 | 100 | ✗ | 0.0 | 100 | 72.77% | 71.61% | 69.69% | 66.72% | 56.62% |
| | | | 0.0 | 256 | 69.26% | 68.72% | 66.55% | 63.48% | 52.98% |
| | | | 0.9 | 100 | 73.90% | 73.45% | 72.31% | 68.19% | 55.91% |
| | | | 0.9 | 256 | **73.98%** | **73.85%** | **72.28%** | **68.96%** | **58.72%** |

## A.3 Winning Tickets without Stability

In Figures 10 to 15 we plot the evolution of Lottery Ticket validation accuracy at all studied sparsities. These graphs serve to complement the results from Table 4 in the main paper, and Tables 12 to 16 above. In those tables we highlight accuracies at specific sparsities, which are indicated on the x-axis in the figures. We can see more clearly in these graphs that the more unstable approaches suffer from more significant accuracy degradation throughout the pruning process, even though the dense networks attained with those procedures have superior performances.

For the results found with VGG-16 we notice that the inflection point where the more stable hyperparameters start to outperform unstable hyperparameters occurs at a higher sparsity than for the ResNet models. By analysing the error barrier results from Figure 7, as well as the optimal rewind points from Table 9, we notice that VGG-16 is significantly more unstable to SGD noise than the ResNet models studied. Likely this is either a result of the different overparameterization of VGG-16 vs ResNet-18, or related to the use of skip connections. We leave an exact explanation for this phenomenon for future work.

As mentioned in the main paper, ResNet-34 + TinyImageNet results did not always converge. As such, some of the curves record a significant accuracy drop together with a high standard deviation. This indicates that at that iteration one (or more) random runs did not train and instead remained at random chance (0.5%).

Table 12: Lottery Ticket extracted from **ResNet-18** on **CIFAR-10** performances for different training parameters at different sparsity levels. Winning tickets are underlined.

| Hyperparameters | | | Sparsity | | | | |
|---|---|---|---|---|---|---|---|
| $\mu$ | $b$ | Pretrained | 0.00% | 67.23% | 89.26% | 96.48% | 98.85% |
| 0.0 | 100 | | 94.58% | 94.39% | 93.93% | 92.95% | 91.10% |
| 0.0 | 256 | ✗ | 93.63% | 94.02% | 93.93% | **93.39%** | **92.67%** |
| 0.9 | 100 | | **95.24%** | **94.99%** | **94.39%** | 93.25% | 91.41% |
| 0.9 | 256 | | 95.18% | 94.98% | 94.29% | 93.36% | 91.39% |
| 0.0 | 100 | | 94.62% | 95.06% | **94.95%** | **94.71%** | **93.95%** |
| 0.0 | 256 | ✓ | 93.54% | 93.97% | 93.94% | 93.50% | 92.77% |
| 0.9 | 100 | | **95.30%** | 95.13% | 94.85% | 93.90% | 92.31% |
| 0.9 | 256 | | 95.24% | **95.16%** | 94.55% | 93.57% | 91.86% |

Table 13: Lottery Ticket extracted from **ResNet-18** on **CIFAR-100** performances for different training parameters at different sparsity levels. Winning tickets are underlined.

| Hyperparameters | | | Sparsity | | | | |
|---|---|---|---|---|---|---|---|
| $\mu$ | $b$ | Pretrained | 0.00% | 67.23% | 89.26% | 96.48% | 98.85% |
| 0.0 | 100 | | 76.66% | 75.57% | 73.53% | 70.24% | 66.43% |
| 0.0 | 256 | ✗ | 74.39% | 75.03% | **74.46%** | **72.43%** | **68.65%** |
| 0.9 | 100 | | **78.54%** | **76.60%** | 73.91% | 70.33% | 65.05% |
| 0.9 | 256 | | 77.74% | 76.47% | 74.07% | 70.23% | 65.89% |
| 0.0 | 100 | | 76.57% | 76.76% | 76.23% | 74.23% | **69.99%** |
| 0.0 | 256 | ✓ | 74.41% | 74.62% | 73.54% | 71.76% | 67.93% |
| 0.9 | 100 | | **78.18%** | 77.20% | 74.66% | 70.58% | 65.80% |
| 0.9 | 256 | | 77.71% | **77.57%** | **77.24%** | **74.35%** | 69.21% |

Table 14: Lottery Ticket extracted from **VGG16** on **CIFAR-100** performances for different training parameters at different sparsity levels. Winning tickets are underlined.

| Hyperparameters | | | Sparsity | | | | |
|---|---|---|---|---|---|---|---|
| $\mu$ | $b$ | Pretrained | 0.00% | 67.23% | 89.26% | 96.48% | 98.85% |
| 0.0 | 100 | | 72.77% | 71.60% | 69.81% | 66.37% | 60.91% |
| 0.0 | 256 | ✗ | 69.26% | 69.53% | 68.80% | 67.54% | **65.41%** |
| 0.9 | 100 | | 73.90% | 73.45% | **72.31%** | 68.19% | 55.91% |
| 0.9 | 256 | | **73.98%** | **73.84%** | 72.25% | **69.50%** | 62.52% |
| 0.0 | 100 | | 72.72% | 72.83% | 72.90% | **71.93%** | 64.43% |
| 0.0 | 256 | ✓ | 69.57% | 70.14% | 69.06% | 67.90% | **65.27%** |
| 0.9 | 100 | | **74.05%** | 73.62% | 72.84% | 68.83% | 53.42% |
| 0.9 | 256 | | 73.99% | **74.05%** | **73.23%** | 70.50% | 64.63% |

Table 15: Lottery Ticket extracted from **Swin** models on **TinyImageNet** performances for different training parameters at different sparsity levels. Winning tickets are underlined.

| Hyperparameters | | | Sparsity | | | | |
|---|---|---|---|---|---|---|---|
| $\mu$ | $b$ | Pretrained | 0.00% | 67.23% | 89.26% | 96.48% | 98.85% |
| 0.0 | 100 | | 50.65% | **52.01%** | **51.01%** | 46.07% | 45.55% |
| 0.0 | 256 | ✗ | 49.52% | 50.01% | 48.97% | **48.79%** | **45.84%** |
| 0.9 | 100 | | **52.53%** | 51.89% | 50.02% | 45.53% | 45.36% |
| 0.9 | 256 | | 52.37% | 50.57% | 49.37% | 45.76% | 45.41% |
| 0.0 | 100 | | 50.29% | **52.39%** | **51.72%** | 49.05% | **46.91%** |
| 0.0 | 256 | ✓ | 49.32% | 50.14% | 49.02% | **49.90%** | 46.26% |
| 0.9 | 100 | | 52.21% | 51.69% | 49.95% | 46.79% | 46.11% |
| 0.9 | 256 | | **52.46%** | 51.92% | 50.19% | 47.71% | 46.90% |

Table 16: Lottery Ticket extracted from **ResNet-50** on **ImageNet-100** performances for different training parameters at different sparsity levels. Winning tickets are underlined.

| Hyperparameters | | | Sparsity | | | |
|---|---|---|---|---|---|---|
| $\mu$ | $b$ | Pretrained | 0.00% | 67.23% | 89.26% | 96.48% |
| 0.0 | 100 | | 82.67% | 83.03% | **82.31%** | **81.75%** |
| 0.0 | 256 | ✗ | 80.67% | 81.35% | 80.60% | 80.37% |
| 0.9 | 100 | | **84.94%** | **83.93%** | 79.57% | 73.07% |
| 0.9 | 256 | | 81.83% | 82.26% | 81.93% | 79.52% |
| 0.0 | 100 | | 82.83% | 83.53% | 83.10% | **82.96%** |
| 0.0 | 256 | ✓ | 80.54% | 81.17% | 81.01% | 80.34% |
| 0.9 | 100 | | **84.61%** | **84.93%** | **84.09%** | 81.11% |
| 0.9 | 256 | | 81.97% | 83.40% | 83.39% | 81.07% |

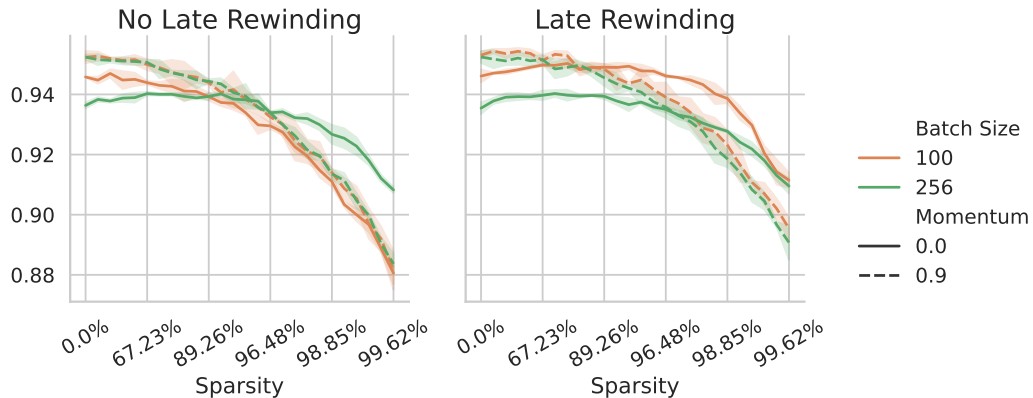

Figure 10: Validation accuracies at different sparsities for Lottery Tickets extracted from a **ResNet-18** model on the **CIFAR-10** dataset.

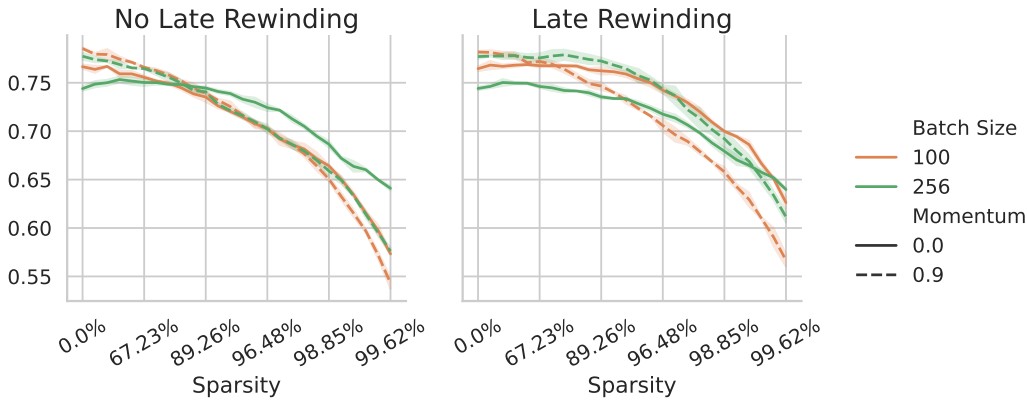

Figure 11: Validation accuracies at different sparsities for Lottery Tickets extracted from a **ResNet-18** model on the **CIFAR-100** dataset.

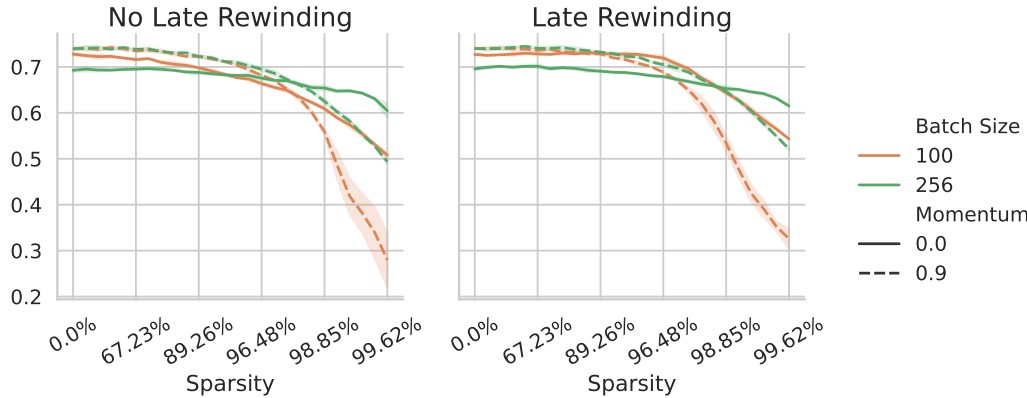

Figure 12: Validation accuracies at different sparsities for Lottery Tickets extracted from a **VGG-16** model on the **CIFAR-100 dataset**.

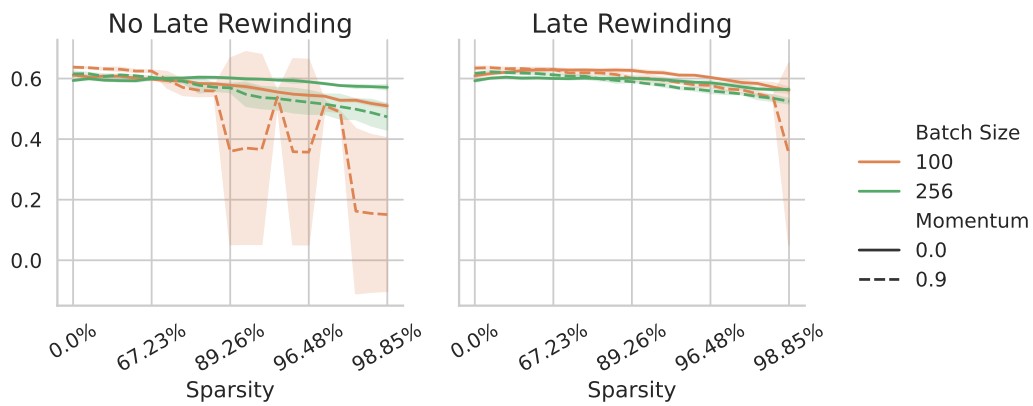

Figure 13: Validation accuracies at different sparsities for Lottery Tickets extracted from a **ResNet-34** model on the **TinyImageNet** dataset.

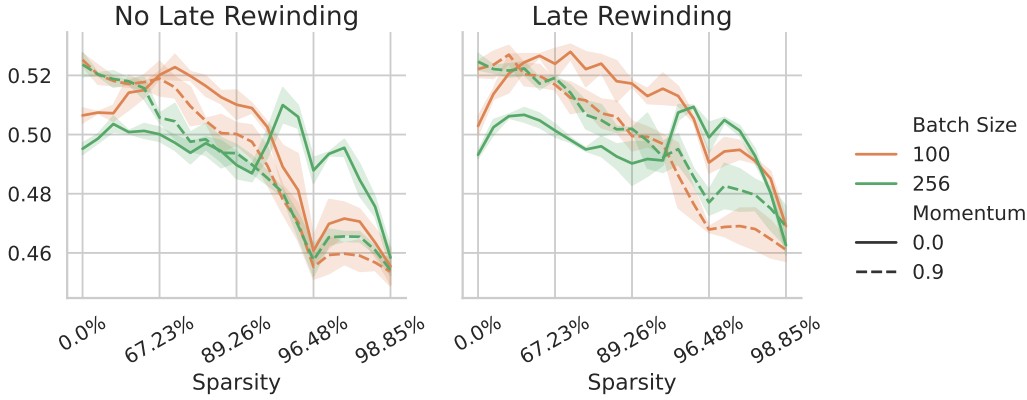

Figure 14: Validation accuracies at different sparsities for Lottery Tickets extracted from a **Swin** model on the **TinyImageNet** dataset.

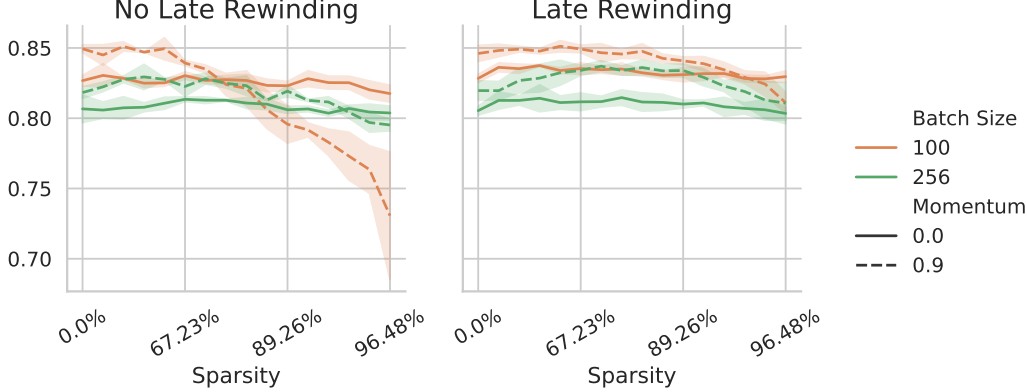

Figure 15: Validation accuracies at different sparsities for Lottery Tickets extracted from a **ResNet-50** model on the **ImageNet-100** dataset.

### A.4 How budget impacts ticket generalization

Extending the analysis in the main paper, we provide two additional settings in Tables 17 and 18 where we use a limited budget, namely ResNet-18 + CIFAR-10 and VGG-16 + CIFAR-100 respectively. Globally, these settings exhibit the same phenomena as observed in ResNet-34 + TinyImageNet where decreasing the training budget aids the generalization performance of late-stable tickets at high sparsities, albeit less exaggerated. Interestingly, in the case of ResNet-18 + CIFAR-10, we can discern a small but noticeable decrease in generalization for the early-stable tickets with a lower budget which is not present in the other settings.

Table 17: AIMP results for different hyperparameter settings of **ResNet-18** on **CIFAR-10**.

| Hyperparameters | | | Sparsity | | | | |
|---|---|---|---|---|---|---|---|
| Budget | $\mu$ | $b$ | 0.00% | 67.23% | 89.26% | 96.48% | 98.85% |
| 20eps | 0.0 | 100 | 94.58% | 94.61% | 94.36% | **93.68%** | **92.32%** |
| | 0.0 | 256 | 93.63% | 93.71% | 92.89% | 92.62% | 90.91% |
| | 0.9 | 100 | **95.24%** | **95.31%** | **94.71%** | 93.62% | 91.65% |
| | 0.9 | 256 | 95.18% | 95.14% | 94.32% | 93.09% | 91.91% |
| 50eps | 0.0 | 100 | 94.58% | 94.46% | 94.29% | **93.90%** | **93.04%** |
| | 0.0 | 256 | 93.63% | 93.91% | 93.69% | 92.98% | 91.54% |
| | 0.9 | 100 | **95.24%** | **95.15%** | 94.30% | 92.89% | 90.84% |
| | 0.9 | 256 | 95.18% | 95.10% | **94.69%** | 93.61% | 91.61% |

Table 18: AIMP results for different hyperparameter settings of **VGG-16** on **CIFAR-100**.

| Hyperparameters | | | Sparsity | | | | |
|---|---|---|---|---|---|---|---|
| Budget | $\mu$ | $b$ | 0.00% | 67.23% | 89.26% | 96.48% | 98.85% |
| 20eps | 0.0 | 100 | 72.77% | 70.83% | 69.46% | 68.18% | 66.68% |
| | 0.0 | 256 | 69.26% | 67.20% | 65.50% | 65.16% | 62.36% |
| | 0.9 | 100 | 73.90% | **_74.02%_** | **72.53%** | 69.38% | 59.70% |
| | 0.9 | 256 | **73.98%** | 73.38% | 71.17% | **69.71%** | **68.35%** |
| 50eps | 0.0 | 100 | 72.77% | 71.49% | 70.44% | 69.24% | **67.96%** |
| | 0.0 | 256 | 69.26% | 67.61% | 66.13% | 64.52% | 64.63% |
| | 0.9 | 100 | 73.90% | **_74.06%_** | 72.61% | 68.48% | 58.14% |
| | 0.9 | 256 | **73.98%** | 73.89% | **72.84%** | **69.76%** | 62.98% |

## A.5 More Hyperparameter configurations

In the main paper, we have demonstrated that lower batch sizes result in a later emergence of stability to SGD noise, but better generalization of the dense network. Conversely, for higher batch sizes, we see a reduction in SGD instability.

**Different batch sizes.** In Table 19, we extend the range of batch sizes tested for ResNet-18 + CIFAR-10 including batch sizes which are typically not used in practice. More specifically, this means using very low batch sizes, such as [8, 32] or very high batch sizes such as [1024, 2048]. We show that smaller batch sizes suffer from significant degradation at higher sparsities, but have higher accuracy at the earlier sparsities. In Figure 16 we clearly see the effect of batch size on instability. We notice that at very large batch sizes, the application of momentum in combination with late rewinding still results in almost stable configurations.

Table 19: Validation accuracy of **ResNet-18** lottery tickets extracted for **CIFAR-10** given more extreme batch sizes at different sparsity levels. CAUTION: these results were attained with only a single run each. Winning tickets are underlined.

| Hyperparameters | | Sparsity | | | | |
|---|---|---|---|---|---|---|
| $\mu$ | $b$ | 0.00% | 67.23% | 89.26% | 96.48% | 98.85% |
| | 8 | $95.29 \pm 0.00\%$ | $\mathbf{95.54 \pm 0.00\%}$ | $\mathbf{94.81 \pm 0.00\%}$ | $\mathbf{93.79 \pm 0.00\%}$ | $\mathbf{91.93 \pm 0.00\%}$ |
| | 32 | $\mathbf{95.41 \pm 0.00\%}$ | $95.29 \pm 0.00\%$ | $94.71 \pm 0.00\%$ | $93.68 \pm 0.00\%$ | $91.72 \pm 0.00\%$ |
| 0.0 | 1024 | $91.87 \pm 0.00\%$ | $91.81 \pm 0.00\%$ | $91.62 \pm 0.00\%$ | $91.24 \pm 0.00\%$ | $89.86 \pm 0.00\%$ |
| | 2048 | $90.92 \pm 0.00\%$ | $90.74 \pm 0.00\%$ | $90.37 \pm 0.00\%$ | $90.00 \pm 0.00\%$ | $88.78 \pm 0.00\%$ |
| | 8 | $92.92 \pm 0.00\%$ | $88.31 \pm 0.00\%$ | $77.92 \pm 0.00\%$ | $70.27 \pm 0.00\%$ | $10.00 \pm 0.00\%$ |
| | 32 | $\mathbf{94.53 \pm 0.00\%}$ | $93.75 \pm 0.00\%$ | $91.49 \pm 0.00\%$ | $90.02 \pm 0.00\%$ | $88.62 \pm 0.00\%$ |
| 0.9 | 1024 | $94.03 \pm 0.00\%$ | $\underline{94.30 \pm 0.00\%}$ | $93.54 \pm 0.00\%$ | $92.80 \pm 0.00\%$ | $90.57 \pm 0.00\%$ |
| | 2048 | $93.14 \pm 0.00\%$ | $\underline{93.46 \pm 0.00\%}$ | $\mathbf{\underline{93.60 \pm 0.00\%}}$ | $\mathbf{\underline{93.54 \pm 0.00\%}}$ | $\mathbf{92.31 \pm 0.00\%}$ |

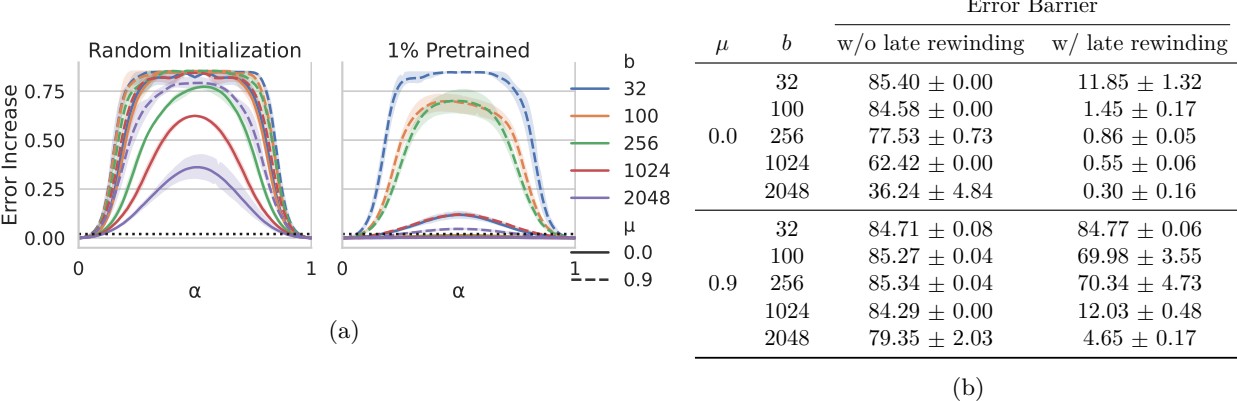

(a)

| | | Error Barrier | |
|---|---|---|---|
| $\mu$ | $b$ | w/o late rewinding | w/ late rewinding |
| | 32 | $85.40 \pm 0.00$ | $11.85 \pm 1.32$ |
| | 100 | $84.58 \pm 0.00$ | $1.45 \pm 0.17$ |
| 0.0 | 256 | $77.53 \pm 0.73$ | $0.86 \pm 0.05$ |
| | 1024 | $62.42 \pm 0.00$ | $0.55 \pm 0.06$ |
| | 2048 | $36.24 \pm 4.84$ | $0.30 \pm 0.16$ |
| | 32 | $84.71 \pm 0.08$ | $84.77 \pm 0.06$ |
| | 100 | $85.27 \pm 0.04$ | $69.98 \pm 3.55$ |
| 0.9 | 256 | $85.34 \pm 0.04$ | $70.34 \pm 4.73$ |
| | 1024 | $84.29 \pm 0.00$ | $12.03 \pm 0.48$ |
| | 2048 | $79.35 \pm 2.03$ | $4.65 \pm 0.17$ |

(b)

Figure 16: Instability for a dense **ResNet-18** model trained on **CIFAR10** with more extreme batch sizes. We show the full interpolation curve in **(a)** and the error barrier in **(b)**.

**Different momentum values.** While it is most common to use $\mu = 0.9$ or $\mu = 0.0$ in training configurations for neural networks, we will explore additional values for this momentum parameter in this section. With this we can provide a more granular overview of the effects on the validation accuracy of the resulting lottery tickets. By comparing the granular momentum results from Table 20 with those found using the common values of $\mu$ in Table 12, we notice that our observations in the main paper also holds true for the more granular values of $\mu$. Indeed, solutions found with a lower value of $\mu$ exhibit lower validation accuracy for the dense network, but have a less severe accuracy drop than those found with higher values of $\mu$. Interestingly the results with $\mu = 0.3$ slightly outperform the results with $\mu = 0.0$, indicating that some small amount of momentum can be beneficial.

Table 20: Validation accuracy of **ResNet-18** lottery tickets extracted for **CIFAR-10** with different values for $\mu$ at different sparsity levels. Winning tickets are underlined.

| Hyperparameters | | Sparsity | | | | |
|---|---|---|---|---|---|---|
| $b$ | $\mu$ | 0.00% | 67.23% | 89.26% | 96.48% | 98.85% |
| | 0.3 | $93.95 \pm 0.36\%$ | $93.92 \pm 0.08\%$ | $\mathbf{\underline{94.07 \pm 0.23\%}}$ | $\mathbf{93.92 \pm 0.27\%}$ | $\mathbf{92.96 \pm 0.17\%}$ |
| 256 | 0.6 | $94.49 \pm 0.23\%$ | $\mathbf{94.31 \pm 0.16\%}$ | $93.76 \pm 0.30\%$ | $92.82 \pm 0.14\%$ | $90.86 \pm 0.17\%$ |
| | 0.95 | $\mathbf{95.15 \pm 0.11\%}$ | $94.29 \pm 0.06\%$ | $93.21 \pm 0.89\%$ | $91.58 \pm 1.12\%$ | $89.89 \pm 1.34\%$ |

### A.6 Learning Rate and Weight Decay

As touched upon in the main paper, other hyperparameters can also impact the existence of winning lottery tickets and their generalization. Here we specifically highlight the impact of the learning rate (lr) and weight decay hyperparameters.

**Learning Rate Scaling.** In an optimization problem the learning rate determines (together with the magnitude of the gradient) the step size the optimizer applies for each update of the weights. Typically a larger learning rate is employed early during training to move towards a low-loss region, and then the optimizer will use a lower learning rate to converge within that loss basin. In this paper we employ cosine annealing for this purpose. However, as the same learning rate is used for all experiments, settings with a lower batch size have more weight updates and thus have more flexibility. In Table 21 we employ the linear scaling rule from (Goyal et al., 2017), where multiplying the batch size by a factor $k$ necessitates multiplying the learning rate by the same factor $k$, to accommodate this and transform from one setting to another. We scale in both directions, defining $\{b = 100, lr = 0.0782\}$ for comparison with $\{b = 256, lr = 0.2\}$ and defining $\{b = 256, lr = 0.512\}$ for comparison with $\{b = 100, lr = 0.2\}$.

We observe that applying this linear scaling rule does not result in comparable validation accuracy with the original settings. Specifically, tickets found with $\{b = 256, lr = 0.512\}$ generalize significantly worse compared to $\{b = 100, lr = 0.2\}$, while the opposite is true for those found with $\{b = 100, lr = 0.0782\}$. This translates into an inversion of the previous trends, where we can now find winning tickets for lower batch sizes, while employing higher batch sizes results in significant accuracy degradation. Importantly, on an even playing field (i.e., if the linear scaling rule is applied in either direction) configurations with $b = 100$ consistently outperform those with $b = 256$.

Table 21: The impact of learning rate on the existence of winning tickets for **ResNet-34** trained on **Tiny-ImageNet**. Starred($^*$) entries encompass one or multiple runs in which the ticket did not converge and remained at random chance.

| Hyperparameters | | | Sparsity | | | | |
|---|---|---|---|---|---|---|---|
| $\mu$ | $b$ | lr | 0.00% | 67.23% | 89.26% | 96.48% | 98.85% |
| 0.0 | 100 | 0.2 | 61.14% | 59.76% | 57.79% | 54.47% | 50.98% |
| | | 0.0782 | 60.27% | 61.05% | 61.10% | 59.54% | 57.79% |
| 0.0 | 256 | 0.2 | 59.33% | 59.95% | 60.20% | 58.89% | 57.09% |
| | | 0.512 | 58.50% | 57.96% | 56.50% | 53.46% | 49.88% |
| 0.9 | 100 | 0.2 | **63.74%** | **62.46%** | 35.95%$^*$ | 35.69%$^*$ | 15.09%$^*$ |
| | | 0.0782 | 62.40% | 61.43% | 58.71% | 55.57% | 50.45% |
| 0.9 | 256 | 0.2 | 61.57% | 60.37% | 56.93% | 52.76% | 47.34% |
| | | 0.512 | 62.48% | 59.93% | 17.62%$^*$ | 34.85%$^*$ | 13.83%$^*$ |

**Weight Decay.** As a form of explicit regularization, weight decay enforces a sense of simplicity on the neural network solution via minimizing the $L_2$ norm of the weights with the aim of improving generalization on unseen data. The application of weight decay is generally associated with the phenomenon of 'grokking' (Liu et al., 2023; Lyu et al., 2024), where generalization on the test dataset emerges significantly later than convergence on the train dataset. In Table 22 we explore other values for this hyperparameter to determine its impact on the generalization of lottery tickets.

We notice that consistently applying a small amount of weight decay results in noticeably better generalizing dense networks. Applying larger values of weight decay can already have a negative effect, but generally still increases the validation accuracy of the dense network. However, in those cases the accuracy degradation experiences by sparse networks is significantly higher than in the settings with limited or no weight decay. This, coupled with the observation that training sparse networks without weight decay does not suffer from

non-convergence, closely follows observations by (Jacobs et al., 2025) that constant weight decay can inhibit the recovery of a sparse ground truth solution.

Table 22: The impact of weight decay on the existence of winning tickets for **ResNet-34** trained on **Tiny-ImageNet**. Starred(*) entries encompass one or multiple runs in which the ticket did not converge and remained at random chance.

| Hyperparameters | | | Sparsity | | | | |
|---|---|---|---|---|---|---|---|
| $\mu$ | $b$ | Weight Decay | 0.00% | 67.23% | 89.26% | 96.48% | 98.85% |
| 0.0 | 100 | | **58.72%** | 58.89% | 57.91% | 55.97% | 53.25% |
| 0.0 | 256 | 0.0 | 58.39% | **60.00%** | **59.53%** | **58.22%** | **55.65%** |
| 0.9 | 100 | | 56.67% | 55.54% | 54.08% | 50.55% | 47.47% |
| 0.9 | 256 | | 54.02% | 55.20% | 53.93% | 50.31% | 47.14% |
| 0.0 | 100 | | 61.14% | 59.76% | 57.79% | 54.47% | 50.98% |
| 0.0 | 256 | 1E-4 | 59.33% | 59.95% | **60.20%** | **58.89%** | **57.09%** |
| 0.9 | 100 | | **63.74%** | **62.46%** | 35.95%* | 35.69%* | 15.09%* |
| 0.9 | 256 | | 61.57% | 60.37% | 56.93% | 52.76% | 47.34% |
| 0.0 | 100 | | **64.66%** | **63.39%** | **60.97%** | **58.07%** | 51.87% |
| 0.0 | 256 | 1E-3 | 62.82% | 61.58% | 60.28% | 57.64% | **52.28%** |
| 0.9 | 100 | | 55.04% | 49.89% | 43.86% | 32.97% | 4.78%* |
| 0.9 | 256 | | 63.66% | 57.60% | 46.60% | 0.50%* | 22.13%* |

## A.7 Mode connectivity between subsequent tickets.

In the main paper we have shown that for ResNet-34 + TinyImageNet late-stable hyperparameters results in winning tickets, while the dense network is not stable to SGD noise. Instead, what happens is that throughout the pruning process the resulting tickets become stable to SGD noise (as seen previously in this section). This has as a result that not only the tickets are stable to SGD noise, but also that the error barriers between the subsequent tickets vanish. Below, in Figures 17 to 21 we demonstrate that this occurrence is not unique to the ResNet-34 + TinyImageNet setting, but rather occurs in all tested settings, albeit the sparsity at which the subsequent tickets lie in the same loss basin differs wildly. The threshold at which this occurs is likely linked to the dataset and model combination as we see different thresholds for different models. Notice that for the Swin + TinyImageNet setting the error barriers between different tickets are markedly higher than in other settings. This is likely due to the use of Attention modules which is not present in other networks.

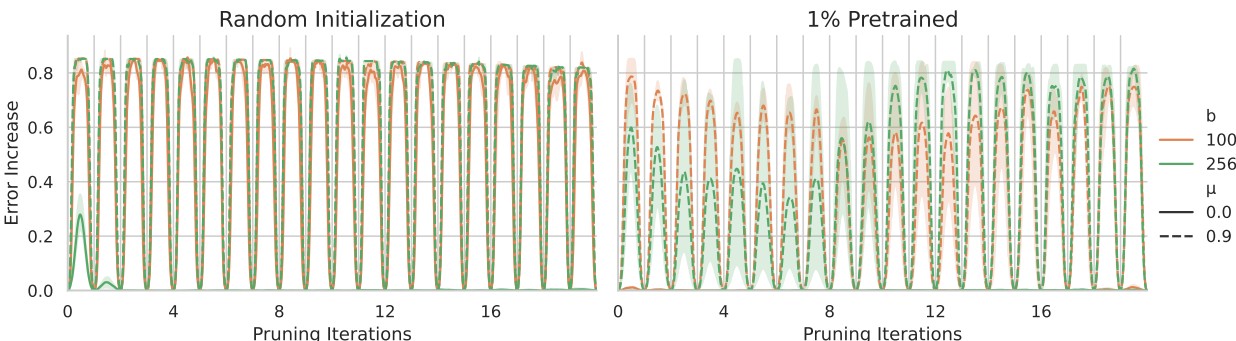

Figure 17: Error barrier between trained tickets found in subsequent pruning iterations for **ResNet-18** trained on **CIFAR-10**. Gray vertical lines indicate the trained tickets.

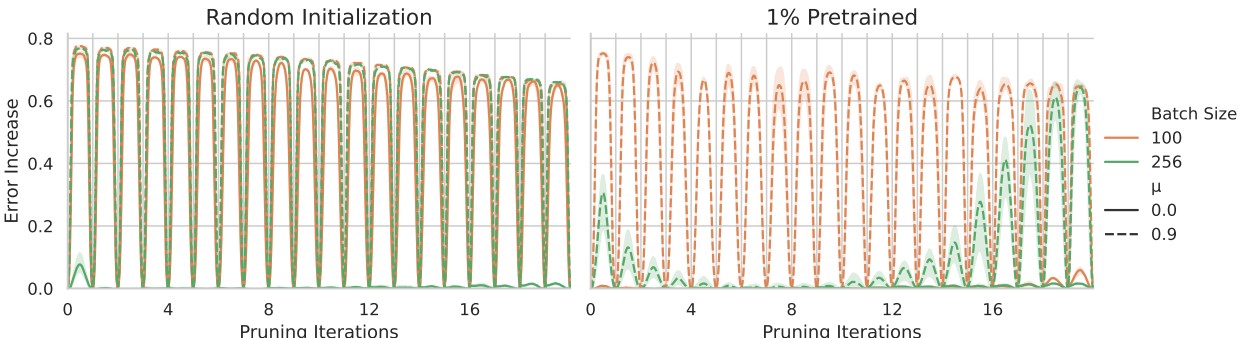

Figure 18: Error barrier between trained tickets found in subsequent pruning iterations for **ResNet-18** trained on **CIFAR-100**. Gray vertical lines indicate the trained tickets.

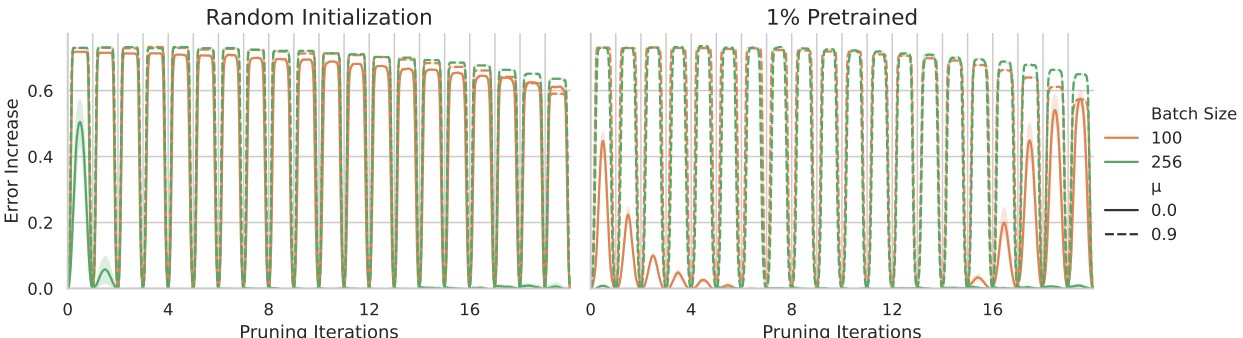

Figure 19: Error barrier between trained tickets found in subsequent pruning iterations for **VGG16** trained on **CIFAR-100**. Gray vertical lines indicate the trained tickets.

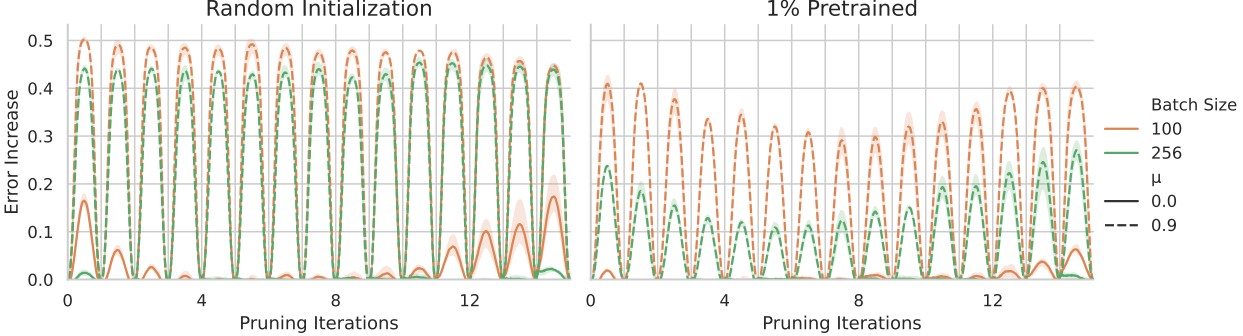

Figure 20: Error barrier between trained tickets found in subsequent pruning iterations for **Swin** trained on **TinyImageNet**. Gray vertical lines indicate the trained tickets.

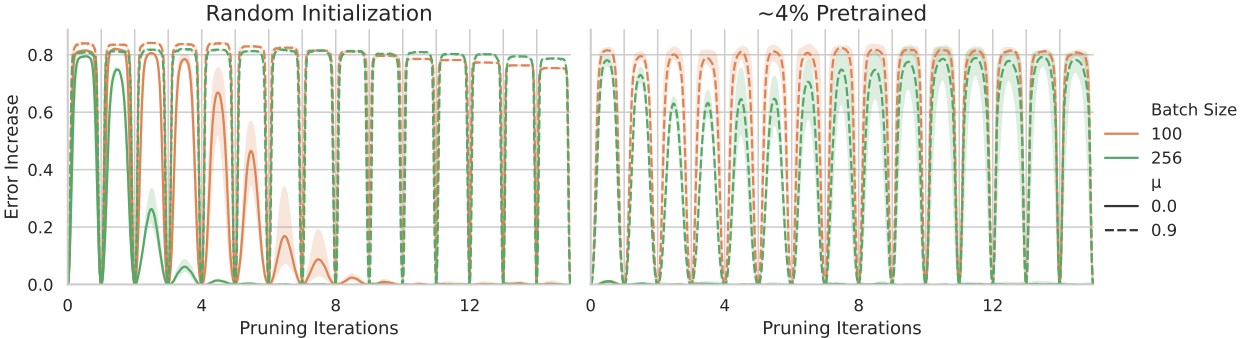

Figure 21: Error barrier between trained tickets found in subsequent pruning iterations for **ResNet-50** trained on **ImageNet100**. Gray vertical lines indicate the trained tickets.

# B  The Emergence of Generalization in Early-stable Tickets

As determined in Appendix A.4 not every dataset network combination is tested with a limited *Mask Search* budget. As such, in the following experiments we list the full budget results for all settings, but show the limited budget results only for the applicable settings (ResNet-18 + CIFAR-10 and VGG-16 + CIFAR-100).

## B.1  Few-shot Generalizability

We provide results for the other considered dataset network combinations in Tables 23 to 27. Opposite to the results for ResNet-34 + TinyImageNet, we see a slight decrease in few-shot generalization when combining the early-stable hyperparameters with a limited *Mask Search* budget. Otherwise, the observations made in the main paper hold for these settings as well.

Table 23: Validation accuracies of a 89.26% sparse **ResNet-18** ticket when trained on a **CIFAR-10** subset. Entries annotated with a † have a sparsity of 0.00%

| Hyperparameters | | | Subset sizes | | | |
|---|---|---|---|---|---|---|
| $\mu$ | $b$ | Budget | 1% | 2% | 5% | 10% |
| 0.0 | 100 | | **87.75 ± 0.49%** | **88.79 ± 0.39%** | **89.95 ± 0.40%** | **90.84 ± 0.15%** |
| 0.0 | 256 | 20eps | 84.19 ± 0.55% | 86.51 ± 0.17% | 88.74 ± 0.05% | 89.48 ± 0.15% |
| 0.9 | 100 | | 39.24 ± 1.51% | 48.48 ± 0.98% | 65.06 ± 0.97% | 78.93 ± 1.47% |
| 0.9 | 256 | | 55.34 ± 5.67% | 63.87 ± 6.16% | 77.71 ± 3.32% | 83.87 ± 1.20% |
| 0.0 | 100 | | 44.22 ± 1.47% | 53.37 ± 1.41% | 68.40 ± 0.96% | 78.90 ± 0.30% |
| 0.0 | 256 | 200eps | **85.43 ± 0.58%** | **88.03 ± 0.62%** | **90.58 ± 0.21%** | **91.82 ± 0.11%** |
| 0.9 | 100 | | 42.93 ± 2.12% | 52.80 ± 3.05% | 71.35 ± 1.49% | 80.96 ± 0.29% |
| 0.9 | 256 | | 37.02 ± 1.03% | 47.38 ± 1.35% | 63.93 ± 4.25% | 75.44 ± 1.05% |
| Permuted Ticket | | | 40.94 ± 1.54% | 50.25 ± 2.77% | 67.28 ± 1.05% | 80.03 ± 0.75% |
| Dense Network † | | | 42.54 ± 1.15% | 51.25 ± 0.78% | 67.15 ± 0.34% | 78.42 ± 0.53% |

Table 24: Validation accuracies of a 89.26% sparse **ResNet-18** ticket when trained on a **CIFAR-100** subset. Entries with a † have a sparsity of 0.00%

| Hyperparameters | | | Subset sizes | | | |
|---|---|---|---|---|---|---|
| $\mu$ | $b$ | Budget | 1% | 2% | 5% | 10% |
| 0.0 | 100 | | $11.20 \pm 0.13\%$ | $16.71 \pm 0.40\%$ | $28.61 \pm 0.40\%$ | $42.01 \pm 0.21\%$ |
| 0.0 | 256 | 200eps | $\mathbf{49.60 \pm 0.53\%}$ | $\mathbf{56.13 \pm 0.84\%}$ | $\mathbf{63.70 \pm 0.26\%}$ | $\mathbf{67.36 \pm 0.44\%}$ |
| 0.9 | 100 | | $8.63 \pm 0.31\%$ | $14.23 \pm 0.42\%$ | $24.13 \pm 0.28\%$ | $37.31 \pm 0.78\%$ |
| 0.9 | 256 | | $9.24 \pm 0.68\%$ | $13.43 \pm 0.61\%$ | $22.54 \pm 1.03\%$ | $35.58 \pm 0.49\%$ |
| Permuted Ticket | | | $9.43 \pm 0.33\%$ | $13.89 \pm 0.25\%$ | $24.70 \pm 1.22\%$ | $35.37 \pm 4.50\%$ |
| Dense Network † | | | $9.44 \pm 0.48\%$ | $13.69 \pm 0.45\%$ | $22.73 \pm 0.09\%$ | $34.34 \pm 0.48\%$ |

Table 25: Validation accuracies of a 89.26% sparse **VGG16** ticket when trained on a **CIFAR-100** subset. Entries with a † have a sparsity of 0.00%

| Hyperparameters | | | Subset sizes | | | |
|---|---|---|---|---|---|---|
| $\mu$ | $b$ | Budget | 1% | 2% | 5% | 10% |
| 0.0 | 100 | | $\mathbf{43.06 \pm 0.95\%}$ | $\mathbf{48.57 \pm 0.59\%}$ | $\mathbf{54.17 \pm 0.21\%}$ | $\mathbf{58.66 \pm 0.19\%}$ |
| 0.0 | 256 | 20eps | $37.13 \pm 0.78\%$ | $43.17 \pm 0.63\%$ | $49.20 \pm 0.39\%$ | $52.56 \pm 0.38\%$ |
| 0.9 | 100 | | $9.12 \pm 0.67\%$ | $13.30 \pm 0.38\%$ | $26.47 \pm 1.59\%$ | $41.37 \pm 0.49\%$ |
| 0.9 | 256 | | $22.04 \pm 1.09\%$ | $33.37 \pm 0.81\%$ | $47.52 \pm 0.78\%$ | $54.95 \pm 0.64\%$ |
| 0.0 | 100 | | $12.07 \pm 0.54\%$ | $16.80 \pm 0.37\%$ | $27.60 \pm 0.63\%$ | $37.82 \pm 0.77\%$ |
| 0.0 | 256 | 200eps | $\mathbf{36.43 \pm 1.11\%}$ | $\mathbf{44.73 \pm 0.07\%}$ | $\mathbf{54.51 \pm 0.15\%}$ | $\mathbf{59.64 \pm 0.19\%}$ |
| 0.9 | 100 | | $7.67 \pm 0.26\%$ | $12.17 \pm 0.18\%$ | $25.03 \pm 1.19\%$ | $40.26 \pm 0.76\%$ |
| 0.9 | 256 | | $7.91 \pm 0.35\%$ | $12.65 \pm 0.83\%$ | $23.81 \pm 0.78\%$ | $38.84 \pm 1.14\%$ |
| Permuted Ticket | | | $9.93 \pm 0.27\%$ | $14.01 \pm 0.51\%$ | $27.24 \pm 0.80\%$ | $42.07 \pm 0.77\%$ |
| Dense Network † | | | $5.25 \pm 0.64\%$ | $9.27 \pm 0.36\%$ | $20.59 \pm 1.01\%$ | $39.26 \pm 0.67\%$ |

Table 26: Validation accuracies of a 89.26% sparse **Swin** ticket when trained on a **TinyImageNet** subset. Entries with a † have a sparsity of 0.00%

| Hyperparameters | | | Subset sizes | | | |
|---|---|---|---|---|---|---|
| $\mu$ | $b$ | Budget | 1% | 2% | 5% | 10% |
| 0.0 | 100 | | $6.01 \pm 0.29\%$ | $11.07 \pm 0.13\%$ | $18.20 \pm 0.32\%$ | $24.25 \pm 0.66\%$ |
| 0.0 | 256 | 200eps | $\mathbf{7.28 \pm 0.17\%}$ | $\mathbf{12.52 \pm 0.49\%}$ | $\mathbf{21.88 \pm 0.47\%}$ | $\mathbf{29.95 \pm 0.19\%}$ |
| 0.9 | 100 | | $5.40 \pm 0.42\%$ | $8.47 \pm 0.24\%$ | $14.69 \pm 0.56\%$ | $20.18 \pm 0.70\%$ |
| 0.9 | 256 | | $5.89 \pm 0.18\%$ | $9.03 \pm 0.48\%$ | $14.65 \pm 0.45\%$ | $20.74 \pm 0.47\%$ |
| Permuted Ticket | | | $6.01 \pm 0.23\%$ | $9.13 \pm 0.20\%$ | $14.94 \pm 0.58\%$ | $20.13 \pm 0.26\%$ |
| Dense Network † | | | $5.47 \pm 0.48\%$ | $7.34 \pm 0.68\%$ | $13.36 \pm 0.46\%$ | $19.51 \pm 0.60\%$ |

Table 27: Validation accuracies of a 89.26% sparse **ResNet-50** ticket when trained on a **ImageNet100** subset. Entries with a † have a sparsity of 0.00%

| Hyperparameters | | | Subset sizes | | | |
|---|---|---|---|---|---|---|
| $\mu$ | $b$ | Budget | 1% | 2% | 5% | 10% |
| 0.0 | 100 | | $37.89 \pm 2.84\%$ | $51.15 \pm 2.50\%$ | $65.67 \pm 0.33\%$ | $72.75 \pm 0.30\%$ |
| 0.0 | 256 | 200eps | $\mathbf{43.80 \pm 3.10\%}$ | $\mathbf{58.81 \pm 0.88\%}$ | $\mathbf{67.06 \pm 0.80\%}$ | $\mathbf{72.76 \pm 0.43\%}$ |
| 0.9 | 100 | | $9.76 \pm 1.32\%$ | $16.45 \pm 1.41\%$ | $30.86 \pm 3.50\%$ | $46.00 \pm 1.71\%$ |
| 0.9 | 256 | | $4.13 \pm 2.06\%$ | $15.47 \pm 1.43\%$ | $30.41 \pm 0.82\%$ | $45.49 \pm 0.12\%$ |
| Permuted Ticket | | | $10.69 \pm 1.51\%$ | $7.03 \pm 9.30\%$ | $11.86 \pm 16.28\%$ | $18.37 \pm 30.09\%$ |
| Dense Network † | | | $4.28 \pm 0.58\%$ | $7.88 \pm 0.48\%$ | $26.77 \pm 3.64\%$ | $52.36 \pm 2.82\%$ |

## B.2 Features encoded in tickets

**Early-stable tickets encode more useful features.** We highlight accuracy for a linear probe trained on the features encoded in a lottery ticket at different sparsities in Tables 28 to 31 for the different dataset-network combinations. We consider the features right before the linear (classification) layer so – depending on the model architecture – these can have different shapes. For the tickets found with the full budget we observe the same results as in the main paper, namely that the early-stable tickets lead to better generalizable probes. However, when studying tickets found with AIMP, we notably observe that these hyperparameters are no longer the best performing, which differs from the ResNet-34 + TinyImageNet setting in the main paper.

**Absence of Swin results.** Interestingly, an outlier is the Swin architecture on TinyImageNet. In that case, the features encoded at initialization in either the dense network or any of the tickets only ever train to random chance, which is why we do not list the results. This is likely due to the bag-of-features approach employed by Transformer models which lack the inherent structural biases of Convolutional Neural Networks (CNNs). We hypothesize that more significant information is encoded in the classification layer, which our approach discards in favor of finetuning one from scratch.

Table 28: **CIFAR-10** validation accuracy for a linear classifier trained on *frozen* **ResNet-18** lottery tickets extracted at different sparsities. Entries annotated with a † have a sparsity of 0.00%.

| Hyperparameters | | | Sparsities | | | |
|---|---|---|---|---|---|---|
| $\mu$ | $b$ | Budget | 67.23% | 89.26% | 96.48% | 98.85% |
| 0.0 | 100 | | $\mathbf{77.77 \pm 1.09\%}$ | $\mathbf{85.85 \pm 0.64\%}$ | $\mathbf{85.41 \pm 0.56\%}$ | $\mathbf{77.37 \pm 0.55\%}$ |
| 0.0 | 256 | 20eps | $68.90 \pm 1.27\%$ | $80.93 \pm 0.68\%$ | $80.55 \pm 0.99\%$ | $68.15 \pm 1.80\%$ |
| 0.9 | 100 | | $28.32 \pm 1.10\%$ | $30.67 \pm 4.08\%$ | $29.32 \pm 1.22\%$ | $32.60 \pm 0.19\%$ |
| 0.9 | 256 | | $43.22 \pm 2.19\%$ | $51.95 \pm 2.99\%$ | $55.13 \pm 5.37\%$ | $46.91 \pm 4.23\%$ |
| 0.0 | 100 | | $29.54 \pm 2.89\%$ | $29.54 \pm 2.53\%$ | $30.84 \pm 0.69\%$ | $30.96 \pm 2.08\%$ |
| 0.0 | 256 | 200eps | $\mathbf{70.48 \pm 1.47\%}$ | $\mathbf{82.87 \pm 1.03\%}$ | $\mathbf{80.85 \pm 1.93\%}$ | $\mathbf{69.64 \pm 2.40\%}$ |
| 0.9 | 100 | | $30.43 \pm 0.78\%$ | $28.65 \pm 1.13\%$ | $25.71 \pm 1.27\%$ | $22.16 \pm 1.39\%$ |
| 0.9 | 256 | | $27.71 \pm 1.58\%$ | $27.83 \pm 1.90\%$ | $28.15 \pm 0.25\%$ | $28.73 \pm 0.45\%$ |
| Permuted Ticket | | | $30.38 \pm 1.62\%$ | $33.69 \pm 0.56\%$ | $31.81 \pm 0.61\%$ | $10.00 \pm 0.00\%$ |
| Dense Network † | | | | $28.20 \pm 1.52\%$ | | |

Table 29: **CIFAR-100** validation accuracy for a linear classifier trained on *frozen* **ResNet-18** lottery tickets extracted at different sparsities. Entries annotated with a † have a sparsity of 0.00%.

| Hyperparameters | | | Sparsities | | | |
|---|---|---|---|---|---|---|
| $\mu$ | $b$ | Budget | 67.23% | 89.26% | 96.48% | 98.85% |
| 0.0 | 100 | | $9.94 \pm 0.28\%$ | $9.31 \pm 0.76\%$ | $10.51 \pm 0.32\%$ | $10.70 \pm 0.67\%$ |
| 0.0 | 256 | 200eps | $\mathbf{31.81 \pm 1.52\%}$ | $\mathbf{44.26 \pm 0.88\%}$ | $\mathbf{36.57 \pm 0.87\%}$ | $\mathbf{21.46 \pm 0.82\%}$ |
| 0.9 | 100 | | $7.15 \pm 0.37\%$ | $7.14 \pm 0.27\%$ | $7.41 \pm 1.54\%$ | $8.14 \pm 0.68\%$ |
| 0.9 | 256 | | $7.41 \pm 0.44\%$ | $7.84 \pm 0.08\%$ | $8.38 \pm 0.62\%$ | $8.14 \pm 0.68\%$ |
| Permuted Ticket | | | $8.50 \pm 0.18\%$ | $10.13 \pm 0.61\%$ | $4.88 \pm 0.18\%$ | $1.00 \pm 0.00\%$ |
| Dense Network † | | | | $8.92 \pm 0.99\%$ | | |

Table 30: **CIFAR-100** validation accuracy for a linear classifier trained on *frozen* **VGG16** lottery tickets extracted at different sparsities. Entries annotated with a † have a sparsity of 0.00%.

| Hyperparameters | | | Sparsities | | | |
|---|---|---|---|---|---|---|
| $\mu$ | $b$ | Budget | 67.23% | 89.26% | 96.48% | 98.85% |
| 0.0 | 100 | | $\mathbf{25.40 \pm 0.75\%}$ | $\mathbf{27.05 \pm 1.24\%}$ | $\mathbf{25.62 \pm 0.42\%}$ | $\mathbf{23.21 \pm 0.50\%}$ |
| 0.0 | 256 | 20eps | $18.30 \pm 0.62\%$ | $19.43 \pm 0.55\%$ | $18.65 \pm 0.62\%$ | $19.80 \pm 0.75\%$ |
| 0.9 | 100 | | $10.24 \pm 1.14\%$ | $12.01 \pm 0.25\%$ | $11.21 \pm 0.50\%$ | $11.20 \pm 0.15\%$ |
| 0.9 | 256 | | $16.23 \pm 0.55\%$ | $21.47 \pm 0.50\%$ | $19.88 \pm 0.96\%$ | $16.20 \pm 0.69\%$ |
| 0.0 | 100 | | $9.62 \pm 0.41\%$ | $9.99 \pm 0.20\%$ | $10.51 \pm 0.16\%$ | $11.53 \pm 0.22\%$ |
| 0.0 | 256 | 200eps | $\mathbf{16.67 \pm 0.47\%}$ | $\mathbf{17.73 \pm 1.32\%}$ | $\mathbf{16.71 \pm 0.42\%}$ | $\mathbf{12.47 \pm 0.37\%}$ |
| 0.9 | 100 | | $8.53 \pm 0.83\%$ | $8.33 \pm 0.74\%$ | $8.49 \pm 0.28\%$ | $6.19 \pm 0.11\%$ |
| 0.9 | 256 | | $10.46 \pm 0.94\%$ | $11.39 \pm 0.49\%$ | $11.27 \pm 0.93\%$ | $11.41 \pm 0.49\%$ |
| Permuted Ticket | | | $8.57 \pm 0.98\%$ | $8.41 \pm 1.01\%$ | $8.56 \pm 0.16\%$ | $6.13 \pm 0.07\%$ |
| Dense Network † | | | | $10.88 \pm 0.75\%$ | | |

Table 31: **ImageNet100** validation accuracy for a linear classifier trained on *frozen* **ResNet-50** lottery tickets extracted at different sparsities. Entries annotated with a † have a sparsity of 0.00%.

| Hyperparameters | | | Sparsities | | |
|---|---|---|---|---|---|
| $\mu$ | $b$ | Budget | 67.23% | 89.26% | 96.48% |
| 0.0 | 100 | | $8.39 \pm 0.12\%$ | $12.71 \pm 0.82\%$ | $12.61 \pm 0.20\%$ |
| 0.0 | 256 | 200eps | $\mathbf{11.72 \pm 0.12\%}$ | $\mathbf{14.66 \pm 1.01\%}$ | $\mathbf{15.09 \pm 0.41\%}$ |
| 0.9 | 100 | | $9.15 \pm 1.63\%$ | $6.64 \pm 0.90\%$ | $5.93 \pm 1.35\%$ |
| 0.9 | 256 | | $9.55 \pm 0.25\%$ | $9.59 \pm 0.79\%$ | $9.42 \pm 0.21\%$ |
| Permuted Ticket | | | $8.64 \pm 0.67\%$ | $6.77 \pm 1.25\%$ | $5.77 \pm 1.24\%$ |
| Dense Network † | | | | $9.90 \pm 0.74\%$ | |

### B.3 Transferability

**Target datasets.** In the main paper, we have transferred from TinyImageNet to {EuroSAT, CIFAR-10, CIFAR-100, CUB-200}. We generally keep the same target datasets for the other settings, but adjust for the relative difference in difficulty between source and target dataset. If the source dataset is CIFAR-10 or CIFAR-100, CUB-200 is replaced by the easier MNIST. In the case of ResNet-50 + ImageNet100 we replace

EuroSAT with the more complex TinyImageNet. When a size mismatch occurs between the target dataset and the expected input size of the network, we rescale the input to resolve this.

**Results.** The results for the different datasets and network combinations are listed in Tables 32 to 36. As noted in the main paper, these follow the trends of the linear evaluation experiments, showing that the features which generalize better to the original dataset, are not specific for that dataset, but also transfer better to other target datasets. In some cases, e.g. transferring from ResNet-18 + CIFAR-10/100 to most datasets, these features can outperform those found in a dense network trained to convergence on the original dataset. The reasoning behind this is not further explored.

Table 32: Transferability to different datasets for frozen 89.26% **ResNet-18** tickets extracted on **CIFAR-10** with different configurations. Entries annotated with a † have a sparsity of 0.00%.

| Hyperparameters | | | Target dataset | | | |
|---|---|---|---|---|---|---|
| $\mu$ | $b$ | Budget | MNIST | EuroSAT | CIFAR-100 | TinyImageNet |
| 0.0 | 100 | | **97.92 ± 0.08%** | **84.62 ± 0.71%** | **42.88 ± 0.81%** | **18.29 ± 1.23%** |
| 0.0 | 256 | 20eps | 96.83 ± 0.26% | 82.77 ± 0.48% | 39.66 ± 1.08% | 17.68 ± 0.83% |
| 0.9 | 100 | | 81.07 ± 0.26% | 36.73 ± 0.77% | 15.60 ± 0.34% | 7.29 ± 0.13% |
| 0.9 | 256 | | 94.59 ± 1.78% | 77.61 ± 3.82% | 25.42 ± 2.02% | 13.63 ± 0.45% |
| 0.0 | 100 | | 82.11 ± 0.48% | 70.46 ± 0.94% | 16.66 ± 0.20% | 7.42 ± 0.14% |
| 0.0 | 256 | 200eps | **97.25 ± 0.15%** | **85.21 ± 0.56%** | **40.57 ± 1.07%** | **18.13 ± 0.33%** |
| 0.9 | 100 | | 77.03 ± 0.31% | 61.00 ± 1.81% | 12.73 ± 0.82% | 5.49 ± 0.21% |
| 0.9 | 256 | | 77.30 ± 1.60% | 60.74 ± 5.21% | 13.35 ± 0.74% | 6.28 ± 0.92% |
| Permuted Ticket | | | 81.66 ± 0.68% | 70.98 ± 1.08% | 15.63 ± 0.13% | 6.29 ± 0.30% |
| *Trained* Dense Network † | | | 71.17 ± 0.81% | 56.42 ± 3.54% | 19.15 ± 0.22% | 4.41 ± 0.25% |

Table 33: Transferability to different datasets for frozen 89.26% **ResNet-18** tickets extracted on **CIFAR-100** with different configurations. Entries annotated with a † have a sparsity of 0.00%.

| Hyperparameters | | | Target dataset | | | |
|---|---|---|---|---|---|---|
| $\mu$ | $b$ | Budget | MNIST | EuroSAT | CIFAR-10 | TinyImageNet |
| 0.0 | 100 | | 84.47 ± 0.80% | 73.05 ± 1.00% | 38.17 ± 0.23% | 8.74 ± 0.31% |
| 0.0 | 256 | 200eps | **97.18 ± 0.20%** | **86.59 ± 0.54%** | **67.60 ± 0.89%** | **20.59 ± 0.97%** |
| 0.9 | 100 | | 79.71 ± 0.79% | 66.08 ± 1.99% | 34.95 ± 1.95% | 6.78 ± 0.07% |
| 0.9 | 256 | | 79.34 ± 3.08% | 68.76 ± 0.70% | 35.84 ± 0.97% | 7.29 ± 0.61% |
| Permuted Ticket | | | 81.35 ± 1.75% | 69.66 ± 0.92% | 35.87 ± 0.70% | 5.96 ± 0.12% |
| *Trained* Dense Network † | | | 90.10 ± 0.36% | 67.04 ± 3.70% | 75.39 ± 0.41% | 13.16 ± 0.46% |

Table 34: Transferability to different datasets for frozen 89.26% **VGG16** tickets extracted on **CIFAR-100** with different configurations. Entries annotated with a † have a sparsity of 0.00%.

| Hyperparameters | | | Target dataset | | | |
|---|---|---|---|---|---|---|
| $\mu$ | $b$ | Budget | MNIST | EuroSAT | CIFAR-10 | TinyImageNet |
| 0.0 | 100 | | **71.50 ± 1.83%** | **60.61 ± 1.82%** | **40.97 ± 0.73%** | **5.08 ± 0.47%** |
| 0.0 | 256 | 20eps | 65.03 ± 1.67% | 54.02 ± 1.56% | 38.55 ± 0.70% | 4.72 ± 0.30% |
| 0.9 | 100 | | 24.18 ± 2.89% | 20.47 ± 2.91% | 15.62 ± 2.80% | 0.70 ± 0.15% |
| 0.9 | 256 | | 64.83 ± 2.38% | 51.25 ± 5.11% | 37.84 ± 0.64% | 3.76 ± 0.14% |
| 0.0 | 100 | | 56.43 ± 4.11% | 52.81 ± 2.62% | 22.84 ± 1.72% | 3.11 ± 0.15% |
| 0.0 | 256 | 200eps | **67.31 ± 1.32%** | **56.78 ± 1.60%** | **35.59 ± 0.51%** | **4.25 ± 0.43%** |
| 0.9 | 100 | | 13.35 ± 2.61% | 31.61 ± 0.95% | 12.45 ± 0.87% | 1.63 ± 0.27% |
| 0.9 | 256 | | 14.52 ± 1.99% | 34.39 ± 3.60% | 14.52 ± 1.99% | 1.51 ± 0.19% |
| Permuted Ticket | | | 11.80 ± 0.09% | 31.89 ± 4.04% | 12.03 ± 0.23% | 1.29 ± 0.13% |
| *Trained* Dense Network † | | | 63.22 ± 0.96% | 48.74 ± 3.69% | 57.18 ± 0.42% | 5.08 ± 0.41% |

Table 35: Transferability to different datasets for frozen 89.26% **Swin** tickets extracted on **TinyImageNet** with different configurations. Entries annotated with a † have a sparsity of 0.00%.

| Hyperparameters | | | Target dataset | | | |
|---|---|---|---|---|---|---|
| $\mu$ | $b$ | Budget | EuroSAT | CIFAR-10 | CIFAR-100 | CUB-200 |
| 0.0 | 100 | | 56.83 ± 0.46% | 42.54 ± 0.23% | 16.48 ± 0.46% | 3.69 ± 0.15% |
| 0.0 | 256 | 200eps | **62.57 ± 1.43%** | **44.61 ± 0.37%** | **18.68 ± 0.46%** | **4.42 ± 0.33%** |
| 0.9 | 100 | | 55.57 ± 0.84% | 41.48 ± 0.76% | 16.49 ± 1.70% | 3.37 ± 0.15% |
| 0.9 | 256 | | 55.82 ± 2.81% | 41.75 ± 1.53% | 16.62 ± 1.88% | 3.26 ± 0.19% |
| Permuted Ticket | | | 54.70 ± 0.26% | 41.55 ± 0.24% | 17.69 ± 0.29% | 6.97 ± 0.11% |
| *Trained* Dense Network † | | | 91.81 ± 0.27% | 74.95 ± 0.41% | 52.84 ± 0.12% | 52.57 ± 0.23% |

Table 36: Transferability to different datasets for frozen 89.26% **ResNet-50** tickets extracted on **ImageNet100** with different configurations. Entries annotated with a † have a sparsity of 0.00%.

| Hyperparameters | | | Target dataset | | | |
|---|---|---|---|---|---|---|
| $\mu$ | $b$ | Budget | CIFAR-10 | CIFAR-100 | TinyImageNet | CUB-200 |
| 0.0 | 100 | | 37.36 ± 1.08% | 15.31 ± 0.04% | 10.37 ± 0.47% | 8.25 ± 0.25% |
| 0.0 | 256 | 200eps | **45.35 ± 0.76%** | **21.96 ± 0.94%** | **18.26 ± 1.03%** | **13.36 ± 0.76%** |
| 0.9 | 100 | | 20.45 ± 1.35% | 4.98 ± 0.68% | 2.38 ± 0.27% | 2.78 ± 0.55% |
| 0.9 | 256 | | 21.55 ± 0.72% | 5.70 ± 0.48% | 2.65 ± 0.33% | 2.85 ± 0.30% |
| Permuted Ticket | | | 23.69 ± 1.30% | 6.78 ± 0.77% | 2.89 ± 0.23% | 3.11 ± 0.25% |
| *Trained* Dense Network † | | | 53.88 ± 1.34% | 31.65 ± 1.00% | 33.57 ± 0.30% | 33.88 ± 0.24% |

# C  Finding Tickets near the Initialization

## C.1  Hyperparameter impact on Learning Distance

In Figure 25 we show the learning distance for the remaining dataset and network combinations. To further highlight the impact of the hyperparameters on this metric, we also show in Figure 22 the learning distances

for the extended hyperparameter configurations studied in Appendix A.5. We also show the impact of lr scaling and weight decay as discussed in Appendix A.6 on the learning distance in Figures 23 and 24 respectively.

**Lr impact.** We can clearly see in Figure 23 that by applying lr rescaling, the learning distance is modified significantly. Indeed, when scaling the lr for settings with $b = 100$, the resulting learning distance aligns almost perfectly for settings with $\{b = 256, lr = 0.2\}$ and vice-versa. This indicates that the interplay of learning rate and batch size determines the distance from the initialization, and helps explain the results listed in Table 21 of Appendix A.6.

**Weight decay impact.** Weight decay imposes an L2 regularization on the weights, preventing them from growing too large, effectively serving as a different kind of limit on the learning distance. While this results in solutions found significantly closer to the initialization (see Figure 24), its application significantly hinders the generalization of sparse tickets (compare Table 22 with Figure 4a). This dichotomy between two close but different regularization targets indicates that learning distance is not the only factor in play.

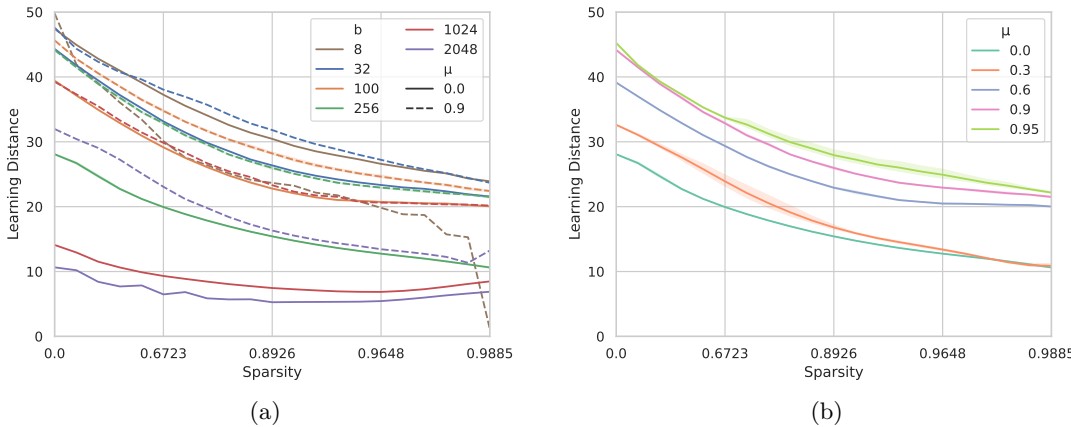

(a)                (b)

Figure 22: Learning distance in the *Mask Search* phase for **ResNet-18** on **CIFAR10** for more diverse **(a)** Batch Sizes and **(b)** Momentum values.

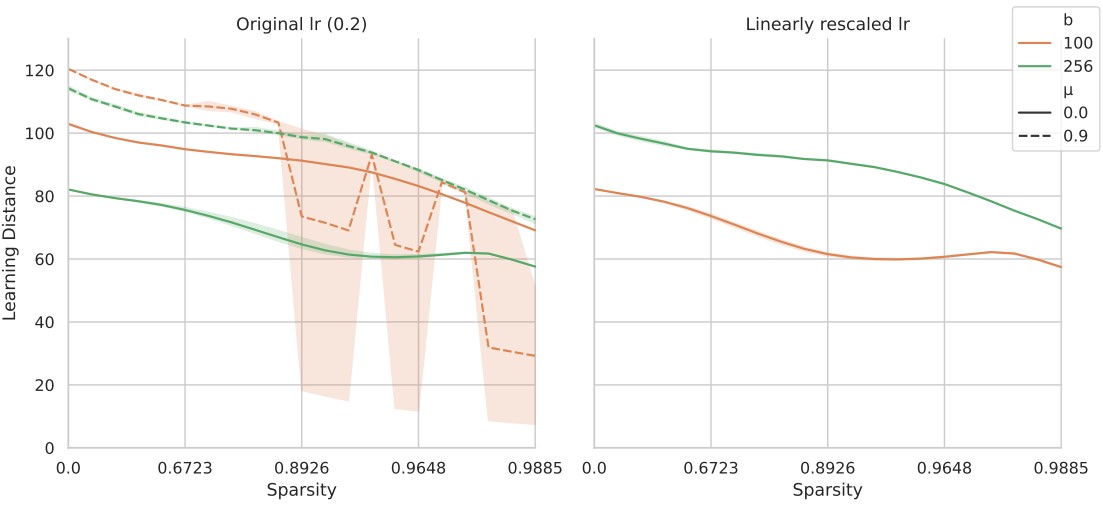

Figure 23: Impact of lr rescaling on the learning distance in the *Mask Search* phase for **ResNet-34** on **TinyImageNet**.

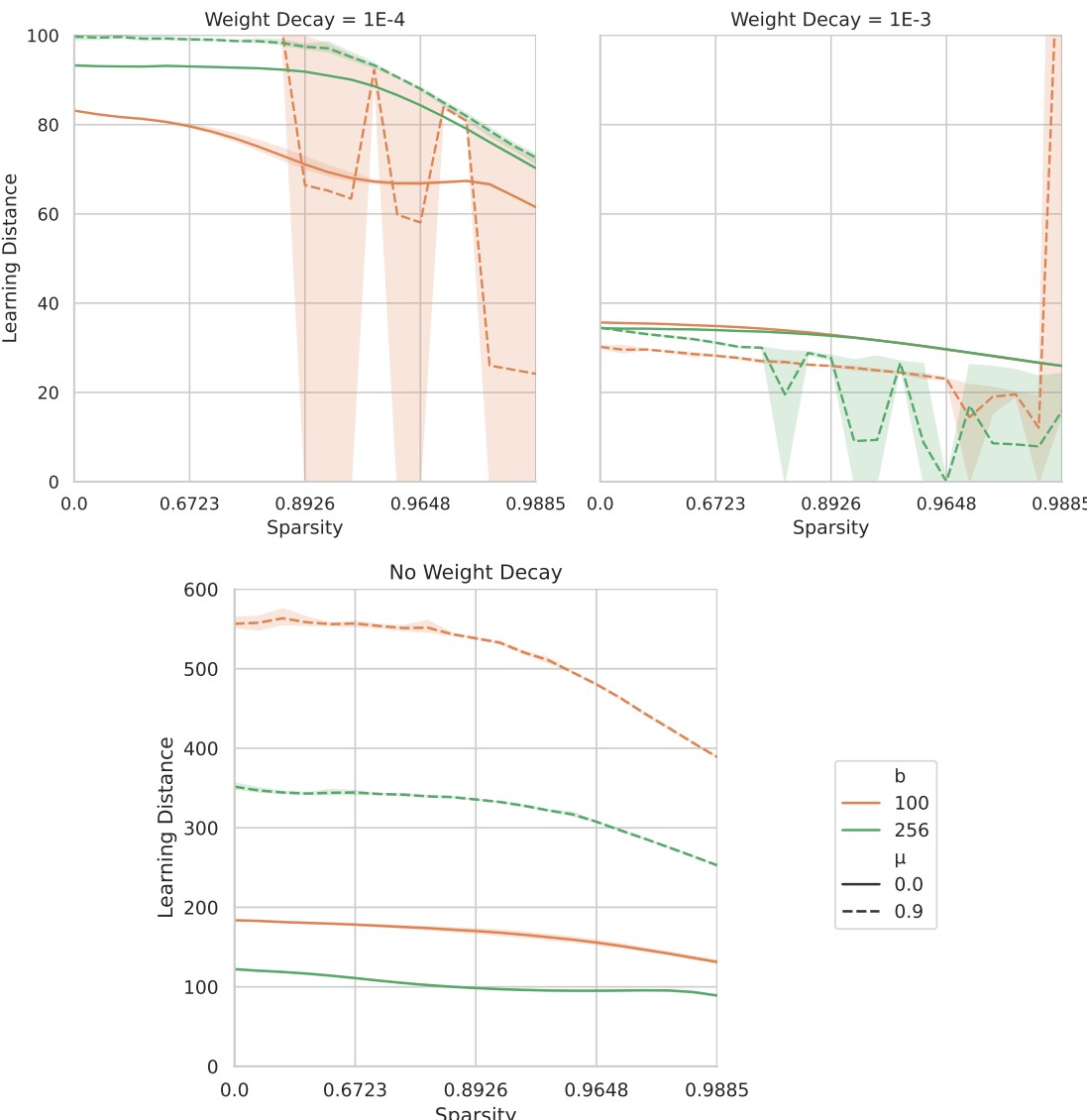

Figure 24: Impact of weight decay on the learning distance in the *Mask Search* phase for **ResNet-34** on **TinyImageNet**.

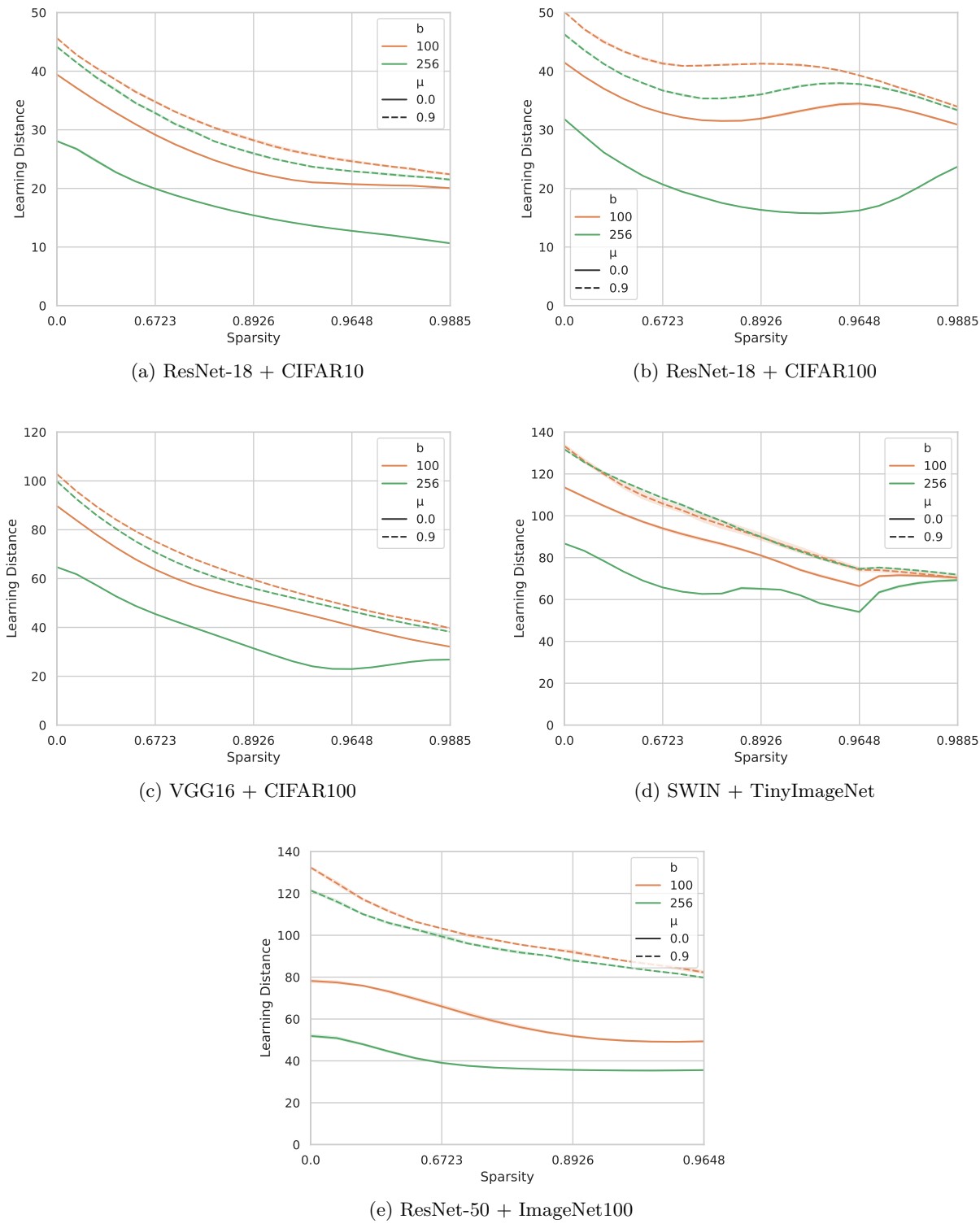

(a) ResNet-18 + CIFAR10

(b) ResNet-18 + CIFAR100

(c) VGG16 + CIFAR100

(d) SWIN + TinyImageNet

(e) ResNet-50 + ImageNet100

Figure 25: Learning distance in the *Mask Search* phase given different hyperparameter settings for various dataset & network combinations without pretraining.

## C.2 Regularizing with Learning Distance

In Figures 26 and 27 we show training with different strengths of learning distance regularization for ResNet-18 + CIFAR10, VGG-16 + CIFAR100 respectively. Notice that the optimal strength differs significantly depending on which dataset-network combination is employed

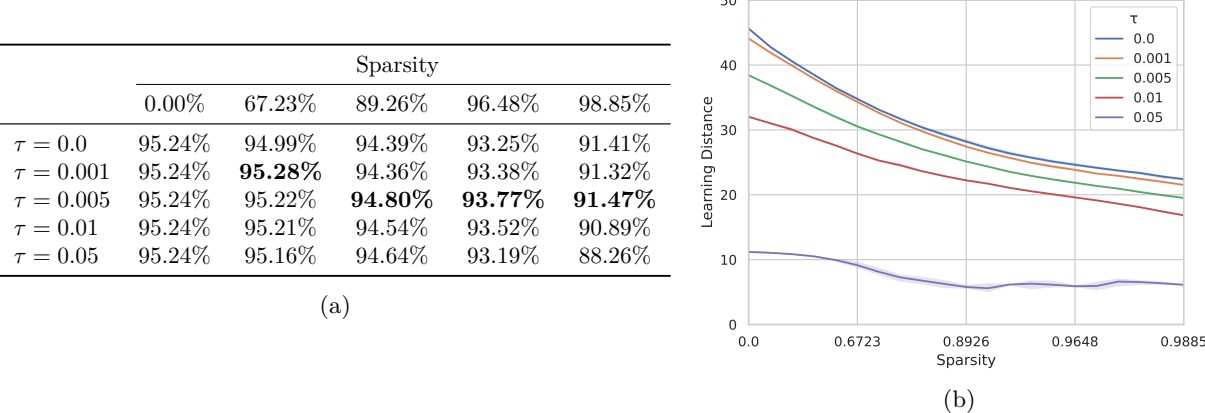

|  | Sparsity | | | | |
| --- | --- | --- | --- | --- | --- |
|  | 0.00% | 67.23% | 89.26% | 96.48% | 98.85% |
| $\tau = 0.0$ | 95.24% | 94.99% | 94.39% | 93.25% | 91.41% |
| $\tau = 0.001$ | 95.24% | **95.28%** | 94.36% | 93.38% | 91.32% |
| $\tau = 0.005$ | 95.24% | 95.22% | **94.80%** | **93.77%** | **91.47%** |
| $\tau = 0.01$ | 95.24% | 95.21% | 94.54% | 93.52% | 90.89% |
| $\tau = 0.05$ | 95.24% | 95.16% | 94.64% | 93.19% | 88.26% |

(a)

(b)

Figure 26: **(a)** Validation accuracy, **(b)** Learning distance for tickets found with **ResNet-18** on **CIFAR-10** ($\{\mu = 0.9,\ b = 100\}$) without pretraining when regularized.

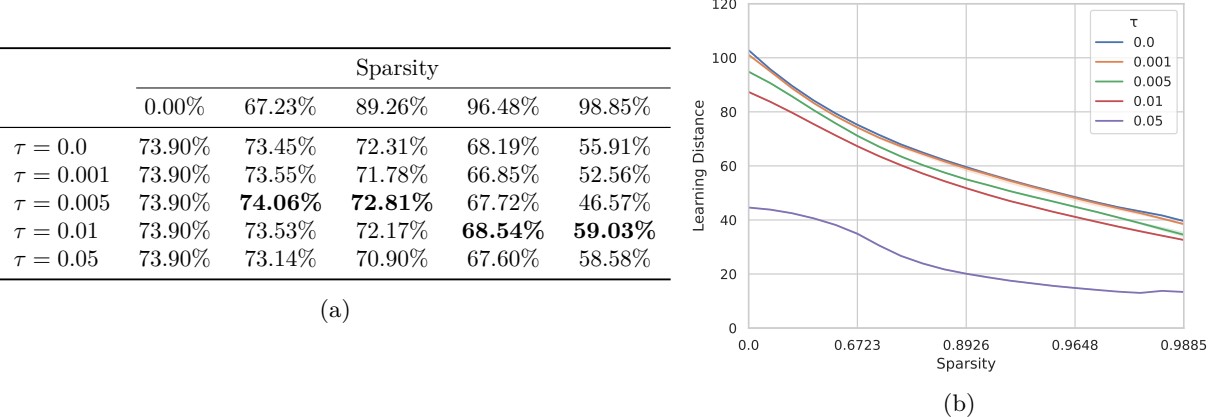

|  | Sparsity | | | | |
| --- | --- | --- | --- | --- | --- |
|  | 0.00% | 67.23% | 89.26% | 96.48% | 98.85% |
| $\tau = 0.0$ | 73.90% | 73.45% | 72.31% | 68.19% | 55.91% |
| $\tau = 0.001$ | 73.90% | 73.55% | 71.78% | 66.85% | 52.56% |
| $\tau = 0.005$ | 73.90% | **74.06%** | **72.81%** | 67.72% | 46.57% |
| $\tau = 0.01$ | 73.90% | 73.53% | 72.17% | **68.54%** | **59.03%** |
| $\tau = 0.05$ | 73.90% | 73.14% | 70.90% | 67.60% | 58.58% |

(a)

(b)

Figure 27: **(a)** Validation accuracy, **(b)** Learning distance for tickets found with **VGG16** on **CIFAR-100** ($\{\mu = 0.9,\ b = 100\}$) without pretraining when regularized.

## D  Hyperparameter configurations

**General hyperparameters.** Each training run contains of 200 (90 for ImageNet100) epochs with a chosen batch size and momentum parameter. Throughout training the learning rate starts at 0.1 (0.2 for TinyImageNet) and is cosine annealed. We also apply a weight decay of 1E-4. Each dataset uses normalization, random cropping and horizontal flipping as data augmentation.

**LTH hyperparameters.** Each iteration 20% of the unpruned parameters are pruned. Late-rewinding involves training for 2 epochs to determine the rewinding point. Each experiments consists of 25 (20 for TinyImageNet, 15 for ImageNet-100) rounds of IMP.

**Linear Probing.** Each linear probe is trained for 5 epochs with a learning rate of 0.005 and weight decay of 1E-4. The probes are trained with $\mu = 0.9$ and $b = 256$. No LR scheduling is used.

**Transferring tickets.** When transferring frozen tickets from one dataset to another, we employ the full training budget, the original learning rate and weight decay, but fix the transfer hyperparameters $\mu = 0.9$ and $b = 256$.

