# OpenReview forum: "Pruning Close to Home: Distance from Initialization impacts Lottery Tickets"
_TMLR — Rejected by TMLR_

### Review · Reviewer_2fx7 · 2026-02-05

**Summary Of Contributions:**

The work investigates the role of important hyperparameters in standard neural network training and how they differently affect lottery ticket training. They show empirically that higher momentum and smaller batchsize lead to worse performance for the lottery tickets training part in contrast to dense training. Moreover, it hypothesized that this is due to parameter distance traveled. This is also supported by experiments on varying the training time. Also this leads to term of early stable configurations.

**Audience:**

Yes

**Audience Explanation:**

Overall most experiments are well motivated and executed and shed light on an important fact for sparse training in general. Therefore, this work can be of interest to the efficiency community and TMLR in general.

**Broader Impact Concerns:**

No concerns, this work is focused on furthering the understanding of deep learning in general.

**Claims And Evidence:**

Yes

**Claims Explanation:**

The main claims regarding the impact of momentum, batch size, and training duration on lottery ticket extraction are generally well substantiated by extensive empirical evidence. The experiments are conducted across multiple architectures and datasets, which strengthens the credibility of the findings. The proposed explanations for the role of each hyperparameter are plausible and consistent with observed training behavior.

However, the scope of the hyperparameter analysis is somewhat limited. Other commonly used hyperparameters (e.g., learning rate schedules, optimizer choice, or regularization strength) may also play a nontrivial role in ticket quality and stability, and exploring at least some of these would have strengthened the claims.

While the paper offers compelling diagnostic insights, it provides limited guidance on how practitioners should act on these findings beyond selecting early-stable configurations. Clearer prescriptions or design principles derived from the analysis would improve the practical clarity of the contributions.

Finally, the evidence in Sections 5.1 and 5.2 is less convincing: the absolute accuracy values of frozen-feature and few-shot experiments are very low overall, making it difficult to assess their practical significance. In addition, several prior works have studied related notions of loss basins, optimization dynamics, and a more explicit discussion of how this work relates to and differs from those perspectives would improve clarity and contextual grounding.

**Requested Changes:**

**Few-shot and frozen-feature experiments (Sections 5.1 and 5.2)**

The experiments in Sections 5.1 and 5.2 are difficult to interpret, as the absolute accuracy values are very low across all settings. This makes it unclear how meaningful the observed differences are in practice. I suggest either removing these sections or replacing them with experiments on a simpler benchmark such as CIFAR-10, following a setup similar to Section 4.1, where differences between configurations are easier to assess.
(Needed)


**Omitted hyperparameters: learning rate and weight decay**

The study focuses on momentum, batch size, and training duration, but does not consider learning rate or weight decay, which are arguably equally important. The learning rate directly controls the extent of feature learning and optimization dynamics (e.g., Tensor Programs [1]), while weight decay induces a simplicity bias that may benefit dense training but harm sparse training [2]. Please either justify why these hyperparameters are excluded or include a limited ablation in one representative setting.
(Needed explanation)


**Relation to recent work on loss basins and optimization dynamics**

Recent work has reported related observations on the optimization challenges of lottery tickets, including analyses based on loss basin structure [3] and parameter sign alignment [4,5]. A more explicit discussion of how the present findings relate to, complement, or differ from these approaches would strengthen the paper’s positioning within the literature.
(Needed)


**Use of ImageNet in the main text**

ImageNet is referenced as a dataset in the main text, but corresponding results are not shown there. Given that ImageNet is a substantially more challenging setting for lottery tickets, it should be clarified why these results are deferred to the appendix or more clearly indicated where they are discussed.
(Improvement of representation)


**Implications beyond vision models**

While the paper focuses on vision architectures trained with SGD and momentum, it would be interesting if the authors could comment on the implications of their findings for other settings, such as modern optimizers or language modeling tasks.
(Out of interest)


**Learning-rate scaling with batch size**

It is unclear whether the learning rate is adapted when varying the batch size. Please clarify whether any form of learning-rate scaling is applied, as this could affect the interpretation of the batch-size results.
(Needed)

[1] Yang, Greg et al. “Tensor Programs V: Tuning Large Neural Networks via Zero-Shot Hyperparameter Transfer.” ArXiv abs/2203.03466 (2022): n. pag.

[2] Jacobs, Tom et al. “Mirror, Mirror of the Flow: How Does Regularization Shape Implicit Bias?” ArXiv abs/2504.12883 (2025): n. pag

[3] Adnan, Mohammed et al. “Sparse Training from Random Initialization: Aligning Lottery Ticket Masks using Weight Symmetry.” ArXiv abs/2505.05143 (2025): n. pag.

[4] Gadhikar, Advait et al. “Sign-In to the Lottery: Reparameterizing Sparse Training From Scratch.” ArXiv abs/2504.12801 (2025): n. pag.

[5] Gadhikar, Advait and Rebekka Burkholz. “Masks, Signs, And Learning Rate Rewinding.” ArXiv abs/2402.19262 (2024): n. pag.

---

> ### Author Response · Authors · 2026-02-23
> **Rebuttal to Reviewer 2fx7**
>
> Dear Reviewer 2fx7,
>
> We thank you for your thorough review. Please find the answers to your questions below.
>
> **Few-shot and frozen-feature experiments (Sections 5.1 and 5.2)**
>
> We actually have experiments with CIFAR-10 and CIFAR-100 in the appendix where indeed the results are significantly more pronounced than those for TinyImageNet. The main reason why those are in the appendix and not in the main paper is because we wanted coherence between the settings for the different experiments in the main text, rather than highlighting the best performances for each setting. Even though the values for TinyImageNet are small in the absolute sense, when compared to those obtained by the different baselines they still encompass a substantial relative increase.
>
> **Omitted hyperparameters: learning rate and weight decay**
>
> Studying different combinations over a larger set of considered hyperparameters would result in a much larger scope of this study which is why we have not conducted such experiments. That being said, indeed lr and weight decay have an impact on the performance after pruning. As such, we have added a simple ablation on ResNet-34 + TinyImageNet for both hyperparameters separately. The results will be included in the appendix of the camera-ready, as these experiments do not fit well in the flow of the current manuscript.
>
> **Relation to recent work on loss basins and optimization dynamics**
>
> We will read and analyze the provided references and strive towards better positioning our work w.r.t. those recent developments.
>
> **Use of ImageNet in the main text**
>
> In the main text we mention the dataset-network combinations used in the whole research. Indeed, we have some experiments for the ImageNet-100 dataset included in the appendix, but not in the main text and not for the full ImageNet dataset. The reason why TinyImageNet is used throughout the main paper is because it provides the right balance of computational complexity, allowing us to conduct large numbers of experiments relatively fast, while still being a less than trivial dataset (as indicated by the validation accuracy of the model). In fact, due to the limited number of samples (500/class), and the large number of classes (200) achieving a good generalization on this dataset proved more difficult than achieving a good generalization on ImageNet-100 with comparably more samples (>=1200/class) and less classes (100).
>
> **Implications beyond vision models**
>
> This is something we have not explored in detail, as our expertise lies squarely in the domain of vision models, similarly to how most literature on the Lottery Ticket Hypothesis is focussed on this domain. We hypothesize that at least for different domains employing the same optimization process via SGD, we could see similar results based on the hyperparameter configurations, but we have no hard proof to stave this hypothesis.
>
> **Learning-rate scaling with batch size**
>
> This is not the case, we employ the same learning rate throughout the different configurations of batch size. Indeed, this can affect the performance as it is typical to employ a larger learning rate with a larger batch size. To accommodate this, we will conduct an additional experiment on ResNet-34 + TinyImageNet where we scale the learning rate depending on the batch size. We follow a linear scaling rule as proposed by [Goyal2018]. This experiment will serve a dual purpose in showing the relationship between lr and batch size on the one hand, and the impact of lr when having a fixed batch size (as discussed earlier in this response).
>
> We have added a new version with the changes indicated in a different color.
>
> Kind regards,
>
> The authors
>
> **References**
>
> [Goyal2018] Goyal et al. 2018 “Accurate, Large Minibatch SGD: Training ImageNet in 1 Hour”

---

> > ### Comment · Reviewer_2fx7 · 2026-02-26
> >
> > Dear Authors,
> >
> > Thank you for answering all of my questions, I consider all questions answered satisfactory.
> > I still would like to mention with respect to my first question, such low accuracy numbers can also be more prone to noise per seed, but that seems not the case, so the significance is true.
> >
> > Kind regards,
> >
> > Reviewer 2fx7

---

### Review · Reviewer_vXzf · 2026-02-24

**Summary Of Contributions:**

Lottery Ticket Hypothesis (LTH) aims to find a sparse subnetwork that can be trained to match the dense model accuracy. However LTH only works with a rewind point for larger models and datasets. This has been attributed to the SGD instability
The paper tries to understand the role of different hyperparameters on the need for rewind point. In particular, the authors studied the role of momentum and batch size on the SGD stability and found momentum ($\mu$) = 0 to be more stable. Based on this insight, the propose to add a regularization term to minimize the learning distance during the mask search.

**Additional Comments:**

What's the intuition behind only studying two hyper params?

Was ViT trained with SGD?

**Audience:**

No

**Audience Explanation:**

The findings of the paper are not clear, results are imcomplete  (sgd noise analysis is done without showing the final accuracy) and some of the results seems to be inaccuracy (at least for now).

**Claims And Evidence:**

No

**Claims Explanation:**

The role of SGD instability has been explored in the literature before, the paper does not explicilty point to out a new insight or observation.  The author claim that the winning tickets can be found without pretraining using $\mu$ = 0 and b.s. = 256 is incorrect as training with this HP results in significantly lower performance.

**Requested Changes:**

Winning ticket is defined by a sparse network that can match the dense accuracy. Most tables and results have not shown the dense model accuracy and where training with a specific HP result in a model that matches the accuracy shown in the literature. Please see more details below.

* Fig. 1: Add the final model accuracy. Without the final acc. the numbers in the table do not add much value.

* > We notice that winning tickets (underlined entries in Table 4) exist in random initializations when using the hyperparameter configuration {𝜇 =0.0,𝑏 = 256}. While this configuration has been deemed unstable to SGD noise (see Figure 1),

   The underlined numbers are clearly not the winning ticket. It is evident from the table that training with $\mu$ = 0 and b.s. = 256 results in lower final accuracy.
While the drop in accuracy between the sparse model (trained without pretraining/rewind) and the dense model is lower with $\mu$ = 0 and b.s. = 256, the overall accuracy of the dense model with this HP is lower than when trained with momentum. The paper consistently makes this error throughout the analysis.

* > Stability can emerge after several Mask Search iterations. Paul et al. (2023) show that if dense networks are stable to SGD noise, then the derived tickets are also stable, and tickets of level 𝐿 and 𝐿+1 can be linearly connected with a low error barrier.

This is technically incorrect. Recent work has shown that all the levels obtained during IMP are linearly mode-connected after taking into account batchnorm [1].

*  Table on page 12 (which is for some reason labelled as fig 4 a) seems to be incorrect. Training without any regularization (i.e. the top row) will correspond to conventional IMP. However the accuracy at 90% and above is too low  from what has been reported in the literature (and I have empirically observed).

[1] Sparse Training from Random Initialization: Aligning Lottery Ticket Masks using Weight Symmetry

---

> ### Author Response · Authors · 2026-03-17
> **Rebuttal to Reviewer vXfz  (1/2)**
>
> Dear reviewer vXzf,
>
> We thank you for your efforts spent reviewing our manuscript. Please find below our rebuttal.
>
> **Mischaracterization of contributions.** The reviewer seems focused on the results in Table 4, regarding the existence of winning tickets for mu=0.0, b=256. They state that no new insights or observations are introduced as SGD instability has been explored before in the literature. Disregarding the fact that no other papers tackle the effect of hyperparameters on SGD noise, meaning that this is already a novel point, we make several other contributions in this manuscript.
>
> Specifically, in Section 5 we show the remarkable generalization present in early-stable tickets for limited data regimes (5.1),  at initialization (5.2), and when transferring to other datasets (5.3). These aspects have not been discussed and studied before in the literature either.
>
> Finally, we introduce the notion that learning distance has an impact on the quality and trainability of the resulting tickets. This is expressed both implicitly in the hyperparameters used which influence this metric and the explicit regularization where we show that applying this explicit regularization can significantly improve the quality of the mask for hyperparameters that otherwise would have suffered rapid accuracy degradation at higher sparsities.
>
> **Definition of winning ticket.** Winning tickets are defined as sparse subnetworks that can match the performance of the dense network when trained under the same circumstances. These include hyperparameters. While indeed in Table 4, tickets found with mu=0.0, b=256 cannot match the accuracy of a dense network trained with mu=0.9, b=100, they can match (and outperform) the dense network found with the same hyperparameters. Furthermore, we base our research upon the observation by [Frankle2020] that network pretraining (and stability to SGD noise) is necessary for winning tickets to exist in more complex settings, which we disprove respectively in Table 4, Figure 2, as well as for other dataset network combinations in the appendix.
>
> **Emergence of stability after several mask search iterations.** The paper you reference [Adnan2025] indeed employs a REPAIR[Jordan2023] technique, which rescales activations to account for batchnorm statistics when interpolating. In that case, they do show that all subsequent iterations in IMP can be linearly connected. In this paper, we follow [Paul2023] and do not use such a technique to modify the interpolants, and rather use default linear interpolation. Regardless of this, we show (1) lower error barriers for a more difficult dataset (albeit after a few pruning iterations), and (2) this linear connectivity also emerges in tickets found from random initialization, while [Adnan2025, Paul2023] both start from pretrained networks which are already stable to SGD noise.
>
> **Accuracies in Table 4a.** As throughout the rest of the paper (except where indicated), we do not use any pretraining to attain the results in this table. Most other papers employ some amount of rewinding to a pretrained initialization as proposed in [Frankle2020] which limits accuracy degradation when pruning more severely.
>
> Furthermore, we list in all our tables the average over three random runs, where in the case of {mu=0.9, b=100/256} we had issues with convergence of some of the sparse networks, not observed in other cases. Rather than discarding these and continuing until we had three runs for each hyperparameter setting, we chose to incorporate this behaviour in the average. This means that these values are significantly lower than if we would only report the converged runs, as the non-converged runs are stuck at random chance (=0.5%). If we would instead disregard these, then the first row of Table 4a would resolve as [63.74%, 62.46%, 53.90%, 53.51%, 45.17%]. This is still significantly worse than the sparse networks found with most of the different values of $\tau$, showing the superiority of regularizing based on learning distance.
>
> **Different hyperparameters.** We limited ourselves to two different hyperparameters in order to be able to conduct extensive experiments without compromising on architectures and datasets. To illustrate this, the results in Table 4 by themselves require ~240 full training runs of a ResNet-34 model on TinyImageNet. Combined with other experiments within the paper, we rather wanted to focus on depth rather than breadth of experiments. In an updated version we have added (limited) experiments on two other hyperparameters based on suggestions made by reviewer 2fx7, namely weight decay and lr, but due to computational limitations these are a lot less extensive than the other experiments.
>
> **ViT Training.** Indeed, the ViT models in this paper are also trained via SGD. While this is not the common approach, we chose to do this because [Frankle2020, Paul2023] focus on stability to SGD noise, rather than another optimizer.
>
> Kind regards,
>
> the authors

---

> > ### Author Response · Authors · 2026-03-17
> > **Rebuttal to reviewer vXfz (2/2)**
> >
> > **References:**
> >
> > [Adnan2025] Adnan et al. 2025 “Sparse Training from Random Initialization: Aligning Lottery Ticket Masks using Weight Symmetry”, ICML’2025
> >
> > [Frankle2020] Frankle et al. “Linear Mode Connectivity and the Lottery Ticket Hypothesis”, ICML’2020
> >
> > [Jordan2023] Jordan et al. “REPAIR: renormalizing permuted activations for interpolation repair.”, ICLR’2023
> >
> > [Paul2023] Paul et al. 2023 “Unmasking the Lottery Ticket Hypothesis: What's Encoded in a Winning Ticket's Mask?”, ICLR’2023

---

> ### Comment · Reviewer_vXzf · 2026-03-19
> **Reply to the authors**
>
> Thank you for your detailed response. However, my concerns remain:
>
> 1. > Definition of winning ticket. Winning tickets are defined as sparse subnetworks that can match the performance of the dense network when trained under the same circumstances.
>
> I disagree with your definition. Your choice of hyper parameters (HP) results in significantly degraded model performance, as other reviewers have noted as well. Though that set of HPs allows you to train without the rewind point, but I think this is an unfair comparison.
>
> Also, pruning at initialization methods yields better accuracy than reported in the paper, which makes it difficult to understand why people should care about the method/analysis proposed in the paper.
>
> While I understand achieving state of the art results is not always the goal for a research project, however I do not think the paper gives a strong insight into Lottery Tickets and why without models trained without momentum exhibit stronger LMC. Reviewr 2fx7 also not the same --- "it provides limited guidance on how practitioners should act on these findings beyond selecting early-stable configurations."
>
>
> 2. >  The paper you reference [Adnan2025] indeed employs a REPAIR[Jordan2023] technique, which rescales activations to account for batchnorm statistics when interpolating.In this paper, we follow [Paul2023] and do not use such a technique to modify the interpolants
>
> Paul et al did not consider the batchnorm stats, perhaps REPAIR was published later. But the correct way to study LMC is by using REPAIR to reset the BN stats.
>
> 3. All the reviewers have pointed out the same --- the paper only consider two hyper parameters, making the study limited in scope.

---

> > ### Author Response · Authors · 2026-03-19
> >
> > Dear reviewer vXzf,
> >
> > To elaborate on your reply:
> >
> > **1.** As an alternative for the early-stable hyperparameters we also explore the regularization with learning distance which provides the same generalization as the 'best' hyperparameters, but does not suffer from such significant accuracy degradation as the case without the regularization. This explicit regularization is a direct continuation of the observation that using early-stable hyperparameters implicitly regularizes the learning distance, so the inclusion of the first part related to hyperparameters is integral to the paper. While PaI methods can possibly achieve better generalization, from my reading of the literature (e.g.,  [Frankle2021]) they typically fail to achieve the same significant sparsity as lottery tickets can, and also act as layerwise sparsity ratios selection, rather than structure selection as lottery tickets do).
> >
> > **2.** We have conducted a small experiment regenerating the results from figure 2 (the error barrier between subsequent trained tickets) by additionally considering the REPAIRing of the BN statistics (using the official code implementation). In that case we can indeed see that the error barriers are lower (see below for a tabular comparison for the first 5 pruning iterations), but the general trend holds that early-stable tickets become linearly mode-connected without pretraining after several pruning iterations, while others do not. We can add this fact in the appendix as an additional discussion point, but the choice of calculation of linear mode connectivity does not detract from the message we make in this paper.
> >
> > Round  | bs=100, mu = 0 | bs=256, mu =0 | bs = 100, mu=0.9 | bs=256, mu 0.9|
> > |----------|----------------------|----------------------|-------------------------| ---------------------|
> > 0 -> 1 | 52.5 (61.0) | 43.5 (59.0) | 62.3 (63.6) | 59.4 (61.6) |
> > 1 -> 2 | 53.5 (60.7) | 35.6 (58.7) | 62.3 (63.6) | 59.3 (61.0) |
> > 2 -> 3 | 53.5 (60.7) | 27.5 (57.7) | 62.1 (63.1) | 59.3 (61.0) |
> > 3 -> 4 | 53.8 (60.4) | 16.4 (55.0) | 61.7 (62.7) | 59.1 (61.0) |
> > 4 -> 5 | 54.1 (60.0) | 7.6 (29.6) | 61.7 (62.5) | 59.6 (60.7) |
> >
> > *In the above table, the value between parentheses is the error barrier as reported in the paper, while the error barrier attained with REPAIR is shown first.*
> >
> > We can see that REPAIR significantly accelerates the rate at which the error barrier vanishes but only in the cases where it vanishes. In other cases there is also a lower error barrier than without REPAIR, but this effect diminishes as the tickets become more sparse. This can also have an effect on the quantities calculated in table 2, as we stability will likely emerge earlier with this more relaxed definition.
> >
> > **3.** At the behest of reviewer 2fx7 we have already added analysis of additional hyperparameters in the updated manuscript, namely learning rate and weight decay. We have also mentioned this in our rebuttal, but perhaps you have overlooked this?
> >
> > Kind regards,
> >
> > The authors
> >
> > **References**
> >
> > [Frankle2021] : Frankle et al. "Pruning Neural Networks at Initialization: Why Are We Missing the Mark?"

---

### Review · Reviewer_bf4R · 2026-03-05

**Summary Of Contributions:**

This paper revisits the Lottery Ticket Hypothesis by examining the role of hyperparameters and whether SGD stability is necessary. It shows that certain settings, such as low momentum and larger batch sizes, allow winning tickets to be found in randomly initialized networks even when the network starts to be unstable. These early-stable configurations stabilize quickly during training. Tickets found this way generalize better, work well with limited data, encode more useful initial features, and transfer more effectively across datasets. This is explained by a learning distance effect. Early-stable training keeps the solution close to the initialization, resulting in better masks. Explicitly regularizing the learning distance also enhances ticket quality for networks that stabilize later.

**Strengths**
1. Shows that stability isn’t required to find winning tickets, which goes against the conventional view from Frankle et al.
2. Few-shot and linear probing experiments reveal interesting insights about what makes a good pruning mask.
3. The learning distance idea is simple and convincing, and the regularization experiment supports it.
4. Covers multiple architectures and datasets, including a Swin transformer, which adds confidence in the results.

**Weakness**
1. The early-stable idea feels a bit post-hoc. The paper shows a correlation, but the causal link between early stability and mask quality isn’t fully explained.
2. Some claims about transferability seem strong, but the CUB-200 results paint a more mixed picture.
3. The connection between learning distance and winning tickets is presented as a hypothesis with some supporting evidence, but weight decay offers a counterexample (it reduces learning distance but hurts ticket quality), which isn’t fully addressed.
4. Finding the specific early-stable hyperparameters and validating the stability point requires significant computation, which can be more expensive than simply rewinding weights to a later epoch as done in traditional LTH.
5. Early-stable settings, like zero momentum or very large batch sizes, can hurt the performance of the full dense model, and the paper does not fully explore this trade-off.

**Audience:**

Yes

**Audience Explanation:**

The LTH literature is active, and why winning tickets exist remains an open question. This paper adds a practical finding along with a mechanistic idea that learning distance drives mask quality.

**Claims And Evidence:**

Yes

**Claims Explanation:**

**The main claims of the paper are well supported.** Tables 4 and 12 to 16 show that early-stable configurations can produce winning tickets without pretraining across multiple settings, which is the central finding. The few-shot results in Table 6 are particularly prominent. The gap between early-stable and late-stable configurations at 1 percent data is large and consistent. Linear probing results in Table 7 further support the mask quality hypothesis, providing a cleaner test since they remove the effect of training dynamics entirely. Overall, these experiments give convincing evidence that early-stable configurations can identify high-quality winning tickets and that learning distance is an important factor.

**There are, however, areas where the evidence is less clear**. Weight decay, shown in Appendices A.6 and C.1, reduces learning distance as seen in Figure 24 but actually hurts ticket generalization as reported in Table 22. The paper acknowledges this briefly, but it deserves more discussion since it challenges the learning distance explanation. Transferability claims are also mixed. For Swin on TinyImageNet shown in Table 35, early-stable configurations do not stand out, and for ResNet-50 on ImageNet100 shown in Table 36, the trained dense network outperforms all ticket variants. While sparse features sometimes beat dense ones, it is less consistent than the text suggests. Finally, the binary search for stability points assumes a monotone decrease in the error barrier as mentioned in footnote 1, but this assumption is not validated and could fail early in training when dynamics are noisy.

**Requested Changes:**

See weaknesses.

---

> ### Author Response · Authors · 2026-03-17
> **Rebuttal to reviewer bf4R (1/2)**
>
> Dear reviewer bf4R,
>
> We thank you for your thorough review, please find below our rebuttal.
>
> *The early-stable idea feels a bit post-hoc…*
>
> In [Frankle2020] it is determined that linear mode connectivity in the form of stability to SGD noise is required to find winning tickets. In contrast, we show that winning tickets can be found in initializations which are not stable to SGD noise, but are close to being stable, indicating that with the ‘early-stable’ name, as under these specific hyperparameters stability to SGD noise emerges much earlier than under other hyperparameters. As such, while we do not explicitly establish a causal relation between early-stable hyperparameters and mask quality, we establish at least a highly explicit correlation. Furthermore, in Table 3 w.r.t. disentangling the impact of Sparse Training and Mask Search hyperparameters, we establish that while the Sparse Training hyperparameters have some impact on the final generalization of the network, the vast majority of its reported generalization hinges upon the presence of early-stable hyperparameters in the Mask Search phase.
>
>
> *...the CUB-200 results paint a more nuanced picture…*
>
> The reason why the CUB-200 results are more underwhelming compared to those on other datasets can likely be linked to the fine-grained nature of this dataset, which is a characteristic that is not present in the other datasets. To confirm this, we could also calculate the top-5 accuracy of these tickets and compare them with those of the dense network, which will better reflect upon the fine-grained nature of CUB-200. However, the goal of these experiments is not to necessarily outperform trained dense networks, but rather show that this significant transferability is unique to early-stable tickets, making them different from the other sparse networks, where such transferability is not present.
>
> *Finding the specific early-stable hyperparameters requires significant computational effort…*
>
> We would argue the contrary. While we make the effort to calculate the rewind point in order to illustrate the difference between early-stable and late-stable hyperparameters, this is not required. In fact, the early-stable hyperparameters we select in one experiment are valid for almost all of the other tested dataset-network combinations as well. Indeed, there seems to be a relatively straightforward guideline that employing higher batch sizes coupled with no momentum results in better generalizable sparsity masks. Opposite to that, when employing pretraining, the recommendation given in [Frankle2020] is to have a pretrained network stable to SGD noise. To calculate the required number of pretrain iterations then would either involve a trial-and-error approach, such as the one we have employed to generate the earliest stable points in Table 2, or involve taking an estimate of the pretraining required, running the risk that this amount is under-, or overestimated.
>
> *Early-stable settings can hurt the performance of the full model**
>
> Indeed, this is a problem. We see that using such hyperparameters harms the performance of the dense network in several tables of the paper (e.g. Tables 3, 4). As such, we use Table 3 to first disentangle the impact of hyperparameters selection in the Mask Search phase and the Sparse Training phase. We establish there that the early-stable hyperparameters affect the final validation accuracy of the dense (and also sparse models!) when used in the Sparse Training phase, but when used in the Mask Search phase, they improve the trainability of the final mask. As such, we propose that for the best possible accuracy, different Sparse Training hyperparameters should be used than the Mask Search phase.
> As an alternative we also introduce the regularization with the Learning Distance. While the early-stable hyperparameters implicitly affect the learning distance but also the final validation accuracy, the learning distance regularization explicitly targets this metric, while still allowing for better loss landscape exploration with the better hyperparameters. This gives a result that is the best of both worlds, with both good generalization of the dense network, and of the ensuing lottery tickets.
>
> *binary search for stability points assumes a monotone decrease*
>
> Before employing the binary search algorithm we have calculated the error barrier for several hyperparameters in the early epochs of training and found them to behave in such a manner, that allows binary search. Because we specifically calculate the error barrier after every epoch, rather than after every (few) iteration(s) of training, these suffer less from the noisy dynamics described in the early phases of training. We will make specific mention of these two factors in the paper to clarify this assumption made in the calculation of the earliest stable epoch.

---

> > ### Author Response · Authors · 2026-03-17
> > **Rebuttal to reviewer bf4R (2/2)**
> >
> > *Learning Distance vs Weight Decay*
> >
> > We indeed observe that the employment of weight decay also has as a side effect that it limits the learning distance. Exactly why one regularization helps the generalization of the sparse ticket, while the other does not is unfortunately an open question still. As this is an addition made on the recommendation of another reviewer, we also did not have the time to study the impact and differences between both techniques in detail. We do however note that we also saw worse performance for some settings when the regularization factor lambda was set too high when using learning distance regularization, which constrained the optimization process too strongly, which can be the same case for the weight decay regularization. We will strive to add a more nuanced discussion of this topic in the paper, but are of the opinion that fully exploring this difference is best left to future work, as weight decay is not an integral part of the story.
> >
> > *While sparse features sometimes beat dense ones, it is less consistent than the text suggests.*
> >
> > It was not our aim to make this an absolute claim, so we will change the tone of this claim to better reflect upon this. Rather it was our aim to show that early-stable tickets contain actually useful features, which is not the case for other sparse networks (both tickets that are not early-stable and permuted). The dense networks are present to serve as a more realistic baseline for comparison, not as a target to beat.
> >
> > Kind regards,
> >
> > The authors
> >
> > **References:**
> >
> > [Frankle2020] : Frankle et al. “Linear mode connectivity and the lottery ticket hypothesis”, ICML’2020

---

### Decision · Action_Editor_4nW8 · 2026-04-17

**Recommendation:** Reject

**Audience:**

Yes

**Audience Explanation:**

The idea of finding training contexts in which there is early stability when training Lottery Tickets is generally interesting to the TMLR research community, primarily for those interested in sparse training generally and the Lottery Ticket Hyopthesis specifically, and so the motivation itself would satisfy this criteria.

The reviewers generally found the motivation of the work and some of the findings interesting, but with limited applicability as summarized in discussion by Reviewer 2fx7:

> The finding is interesting and provides various new avenues of research regarding sparse training and Lottery Tickets (finding a potential replacement for momentum for example in the presence of sparsity). However, the work could be improved both theoretically and empirically as some results fall short performance wise from the general baseline. This limits the applicability of the insight.

This was consistently raised by the other reviewers in discussion, as summarized by Reviewer Bf4R:

> The core finding (hyperparameter choice affects SGD stability and ticket quality) is an interesting refinement of Frankle et al. (2020) rather than a fundamental new insight.

and  Reviewer bf4R:

> On the practical side, the guidance is somewhat narrow, focusing on just a couple of hyperparameters, and even there the transferability across different settings is inconsistent.

My overall impression from the reviewer's comments is that this paper is borderline on this evaluation criteria, as the paper presented interesting findings, but with limited implications for the community.

**Claims And Evidence:**

No

**Claims Explanation:**

Most of the reviewers' concerns rested on the empirical results presented, especially around the accuracy of the solutions found, and if these represent lottery ticket solutions due to the gap in performance compared to the dense baselines, and lack of inclusion of other common hyper-parameters used during training.

The authors presented some evidence during rebuttal that addressed some of these concerns, but even after rebuttal multiple reviewers did not find the evidence presented clear and convincing, especially in regard to the results concerning the counter-example of weight decay to the authors' learning distance hypothesis, and the performance gap with dense training with typical hyper-parameters for many of the empirical results.

On the latter issue, the authors argued that when comparing dense vs. LTH, like-hyper parameters should be considered, and in this context they see similar performance to the dense model. However, multiple reviewers found it notable that the generalization performance here is considerably lower than standard dense results, severely limiting any practical impact of finding a stable hyper-parameter configuration. In making a decision here, I believe that while the authors make a valid point in matching HP in terms of understanding the optimization issues within that narrow context, if we discount matching the best dense generalization performance as a key objective with lottery ticket training, I believe lottery ticket training loses critical relevance.

In summary, the work presents some interesting findings in the context of understanding stability in lottery ticket training, and authors' rebuttal provided much needed improvements to the paper, but even post-rebuttal I believe it's fair to say that the reviewers overall did not find the evidence clear and convincing, especially in regards to results or insights that are clearly motivated and practicable in training lottery ticket solutions with dense-like generalization performance.